



# Insights into the effect of spatial and temporal flow variations on turbulent heat exchange at a mountain glacier

Rebecca Mott[1,2], Ivana Stipserki[3], Lindsey Nicholson[3], Jordan Mertes[3]

[1]WSL Institute for Snow and Avalanche Research SLF, Davos, Switzerland

[2]Institute of Meteorology and Climate Research, Atmospheric Environmental Research (KIT/IMK-IFU), Garmisch-Partenkirchen, Germany

[3]Department of Atmospheric and Cryospheric Sciences, University of Innsbruck, Innsbruck, Austria

*Correspondence to*: Rebecca Mott (mott@slf.ch)

**Abstract.** Multi-scale interactions between the glacier surface, the overlying atmosphere and the surrounding alpine terrain

are highly complex. The high heterogeneity of boundary layer processes that couple these systems drives temporally and spatially variable energy fluxes and melt rates. A comprehensive measurement campaign, the HEFEX (Hintereisferner Experiment), was conducted during the summer of 2018. The aim of this experiment was to investigate spatial and temporal dynamics of the near-surface boundary layer and associated heat exchange processes close to the glacier surface during the melting season. The experimental setup of five meteorological stations was designed to capture the spatial and temporal

characteristics of the local wind system on the glacier and to quantify the contribution of horizontal heat advection from surrounding ice-free areas to the local energy flux variability at the glacier. Turbulence data suggest that the temporal change in the local wind system strongly affect the micrometeorology at the glacier. Low-level katabatic flows were persistently measured during both night time and daytime and were responsible for consistently low near-surface air temperatures with small spatial variations at the glacier. On the contrary, local turbulence profiles of momentum and heat revealed strong changes

of the local thermodynamic characteristics at the glacier when larger-scale westerly flows disturbed the prevailing katabatic flow forming low-level across-glacier flows. Warm air advection from the surrounding ice-free areas significantly increased near-surface air-temperatures at the glacier, with strong horizontal temperature gradients from the peripheral zones towards the centerline of the glacier. Despite generally lower near-surface wind speeds during the across-glacier flow, peak horizontal heat advection from the peripheral zones towards the centerline and strong transport of turbulence from higher atmospheric

layers downward resulted in enhanced turbulent heat exchange towards the glacier surface at the glacier centerline. Thus, at the centerline of the glacier the exposure to strong larger-scale westerly winds promoted heat exchange processes at the glacier surface potentially contributing to ice melt. On the contrary, at the peripheral zones of the glacier turbulence data indicate that stronger sheltering from the larger-scale flows allowed the preservation of a katabatic jet, which suppressed the efficiency of



the across-glacier flow to drive heat exchange towards the glacier surface by decoupling low-level atmospheric layers from

the flow aloft. To explain the origin of the across-glacier flow would however require large eddy simulations.

## 1 Introduction

Mountain glaciers are important contributors to the regional and global hydrological cycle (e.g., Bahr and Radić, 2012) as well as sea-level rise (e.g., Radić and Hock, 2011). Thus, it is crucial to understand their mass balance and its climatic drivers. Winter precipitation, avalanching (e.g., Kuhn, 1995; Sold et al., 2013; Mott et al., 2019), wind transport (e.g., Dadic et al.,

2010), regional climate (e.g., Kaser et al., 2004) and micrometeorology (e.g., Kuhn, 1985; Denby and Greuell, 2000; Escher-Vetter, 2002; Oerlemans and Van Den Broeke, 2002; Strasser et al., 2004; Nicholson et al., 2013; Petersen et al., 2013; Conway and Cullen, 2016; Mott et al., 2019) have been found to be driving factors for the survival of mountain glaciers, in the face of generally increasingly unfavorable conditions. The specific contribution of various climatic drivers to the prevalent rapid mass losses of mountain glaciers has been studied using energy balance models (e.g. Mölg et al., 2009; Klok and Oerlemans, 2002).

Although shortwave radiation is the main driver for snow and ice melt, the sensitivity of the melt rate to temperature is controlled by the net longwave radiation and the turbulent heat flux (e.g. Oerlemans, 2001; Cullen and Conway, 2015). Recent studies could, however demonstrate insufficient representation of the variability of energy fluxes on mountain glaciers (e.g. MacDougall and Flowers, 2011; Prinz et al., 2016; Sauter and Galos, 2016). Potentially large bias in snowmelt predictions were also shown to be induced by the evolution of small-scale flow systems in alpine catchments (Mott et al., 2015; Dadic et

al., 2013; Helbig et al., 2017; Schlögl et al.,2018a, b). Several studies already highlighted that complex wind systems at glaciers, with strong spatial and temporal variations of the katabatic flow and interactions with cross-valley flows, drive large variations in the local air temperature field (Petersen and Pellicciotti, 2011) and in turbulent heat exchange (Sauter and Galos, 2016). Several studies claim that deep glacier winds act as heat pump for the glacier surface by generating shear and enhancing turbulent mixing close to the glacier surface (Oerlemans and Grisogono, 2002). Near-surface warming induced by katabatic

flow could also be caused by along-slope warm-air advection (Zhong and Whiteman, 2008) or the entrainment of potentially warmer air down to the surface driven by stronger turbulent mixing (Pinto et al., 2006). Other studies highlighted the effect of katabatic flows in laterally decoupling the local atmosphere from its surrounding, thus lowering the climatic sensitivity of glaciers to external temperature changes (Shea and Moore, 2010; Sauter and Galos, 2016; Mott et al., 2019).

The effect of katabatic wind systems on the local air temperatures over glaciers has been intensively studied and

parameterizations for turbulent fluxes have been suggested (e.g., Oerlemans and Grisogono, 2002; Petersen et al., 2013). However, the complex interaction between different boundary layer processes on glacier mass balance has gained little attention so far. Recently, experimental and numerical studies on turbulent fluxes in the stable boundary layer of snow or ice (Daly et al., 2010; Mott et al., 2013; Curtis et al., 2014; Mott et al., 2016; Mott et al., 2017; Lapo et al., 2019) identified cold-air pooling, boundary layer decoupling and advective heat transport as important counteracting processes altering the local air

temperature and heat exchange processes. Advective transport of sensible heat has been shown to increase the local air



temperature, strongly contributing to the net available melt energy for snow and ice (Essery et al., 2006; Mott et al., 2011; Harder et al., 2017; Schlögl et al., 2018a, b). The numerical simulations of Sauter and Galos (2016) showed that insufficient characterization of these temperature advection processes caused incorrect local sensible heat flux estimates. They showed that cross-valley flows in particular strongly drive the advection of warmer air from surrounding ice-free areas towards the glacier. The increase in local air temperatures enhance the turbulent heat exchange towards the glacier surface, particularly at the peripheral zones of the glacier.

The concurrent existence of counteracting processes such as katabatic flows, horizontal warm air advection and boundary layer decoupling increases the complexity of atmospheric boundary layer dynamics on glaciers, and the interaction between them is not well understood. Warm air advection may disturb the katabatic flow at some areas of the glacier altering thermal conditions and enhancing downward heat exchange towards the glacier surface (Ayala et al., 2015). In the presence of advective heat transport, however, shallow internal boundary layers may enhance local atmospheric stratification close to the snow surfaces resulting in atmospheric decoupling of the air adjacent to the snow cover from the warm air above (Mott et al., 2017). The collapse of near-surface turbulence subsequently limits the amount of sensible and latent heat than can be transmitted from the atmosphere to the snow surface (Mott et al., 2018). Understanding the interplay of these processes is important for correctly interpreting the climatic significance of glacier mass balance studies that typically use interpolated fields for turbulent flux estimations.

## 2 Methods

### 2.1 Field site

The Hintereisferner is a valley glacier located in the Ötztal Alps, Austria. It has been classified as one of the 'reference glaciers' by the World Glacier Monitoring Service, with observations dating back to the year 1952/53, and continuing to the present day as part of a comprehensive catchment monitoring program (Strasser et al., 2018). The mass balance of the glacier has been extensively studied for decades (e.g., Hoinkes, 1970; Kuhn et al., 1999; Marzeion et al., 2012). In addition to traditional glaciological mass balance measurements, numerous ALS flight campaigns were carried out near the end of each mass balance year since 2001 (Klug et al., 2018) and has been used for development and testing of instruments, methods and models (Kuhn et al., 1999) and for investigating glacier and valley winds (Obleitner, 1994).

Hintereisferner is a classical valley glacier approximately 6,3 km long (in 2018) with an elevation difference of approximately 1200 m (www.wgms.ch). The glacier tongue is located in a northeast-orientated valley surrounded by steep slopes (Fig. 1b). In the central part of the glacier tongue the Langtaufererjoch-valley discharges into the main valley, marking the former confluence of a tributary glacier. Hintereisferner is located in the "inner dry Alpine zone"(Frei and Schär, 1998), among the driest regions of the entire European Alps. Like many glaciers in the Eastern Alps the Hintereisferner has experienced strong shrinkage during recent decades. Between 2001 and 2011 the area of the glacier decreased by 15 % (Abermann et al., 2009; Klug et al., 2018).



## 2.2 Turbulence towers

The HEFEX micrometeorological measurement campaign was conducted during three weeks in August 2018. Measurement
towers were installed on the 1. and 2. August and removed on 22. August. The measurement network consisted of five 3-m
tripod towers (Fig. 1a), located at an along- and an across-glacier transect to capture the spatial variations of the atmospheric
flow system at the glacier and associated heat exchange processes. Floating tripods were chosen to allow the towers to migrate
with the melting ice surface and maintain the same sensor height over the experiment.

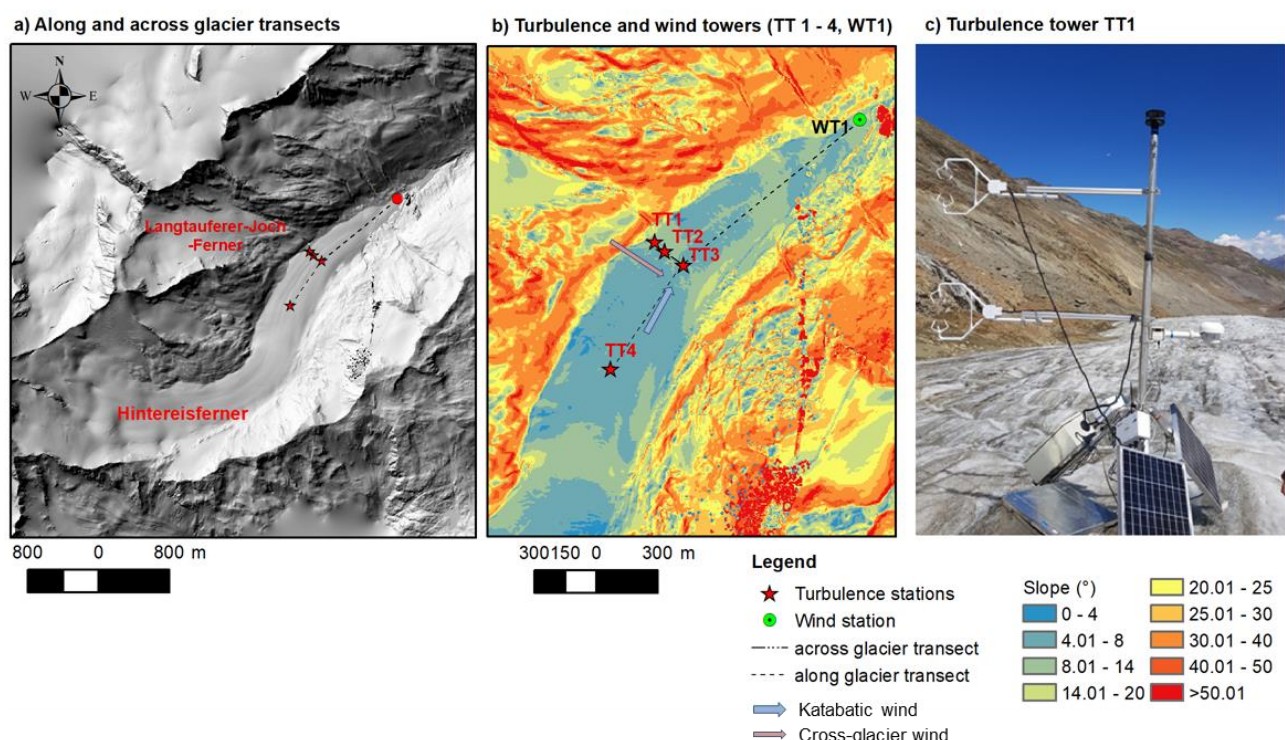


*Figure 1: Experimental test site Hintereisferner with an along- and across-glacier transect of five meteorological towers (a,
b). Four of these towers, the turbulence towers (TT1 – TT4), were additionally equipped with two turbulence sensors (c). The
wind station WT1, installed at the glacier tongue was equipped with three wind sensors. The hillshade (a) and slope maps (b)
were produced based on a terrestrial laser scan of the glacier surface (August 2018), which was combined with an airborne*
*LiDAR scan (September 2013) covering a larger area including the surrounding of the glacier.*

The across-glacier transect consisted of three turbulence towers installed from the peripheral zones of the glacier towards the
centerline (TT1, TT2, TT3) at 2700 m asl (Fig. 1). The location of the across-glacier transect coincides with where the valley
of Langtaufererjochferner discharges into the valley of the Hintereisferner glacier (Fig. 1a). In this area, thermal flows from





the surrounding area were hypothesized to influence the surface of Hintereisferner. The distances between towers TT1 and TT2 were 65 m and 110 m between TT2 and TT3. One turbulence tower (TT4) was installed at an up-glacier location at the glacier centerline (at 2761 m asl), with a horizontal distance of 620 m to TT3. The fifth station (WT1) was installed at the glacier tongue. All stations were installed at comparatively flat areas of the glacier with slope angles varying between 6 and 8°. Measurement towers were installed directly at the ice surface. Due to pronounced changes in the ice surface caused by

strong ice melt during the measurement campaign, frequent visual inspection and small adjustments to the location of the towers were essential for good data quality. This mainly consisted of repositioning the tower feet to ensure the tower stability and re-levelling the sensors. Post-processing of data, i.e. correction of data for height changes and rotation of the mast further ensured data quality (see details below).

Each tower measured wind properties at three heights above the ice surface (1.7 m (level 1) and 2.35 m (level 2) and 2.9 m

(level 3)), as well as air temperature, relative humidity and pressure at level 1. At the four turbulence towers (TT1-TT4) the wind sensors at level 1 and 2 were Campbell CSAT3 sonic anemometers, sampling at a frequency of 20 Hz, while as the fifth tower (WT1), with at these levels was recorded with two Young anemometers. At all towers the level 3 wind sensor was a two-dimensional sonic anemometer. Air temperature, relative humidity and air pressure was measured at each station at measurement level 1 with a 1-minute resolution.

## 2.3 Methodology

The turbulence data were processed as follows: multi-resolution flux decomposition was used to determine the optimal averaging time for the turbulence data (Vickers and Mahrt, 2003). Based on this analysis, the data were block averaged with an averaging time of 1 minute. This short averaging time is necessary to separate turbulence and non-turbulence sub-mesoscale motions in stable boundary layer data, and was also shown to be appropriate over non-glaciated surfaces (Stiperski et al. 2019a,

b). Prior to block averaging the data were rotated using double rotation (Stiperski and Rotach 2016) and detrended with 1-minute temporal resolution (Aubinet, 2012). Double-rotation is preferred over a planar fit method due to continual changes to the surface of the glacier and movement of the stations. The $z$ component is therefore in the local slope normal direction. Data were also corrected for rotation and repositioning of the stations during the campaign caused by strong melting of the glacier surface and associated changes in surface structure of the glacier. Finally, to calculate the advective terms, we rotated the

coordinate system in such a way that $x$ direction is facing down the glacier ($\overline{U} > 0$) and y direction is oriented to the orographic left along the across-glacier transect towards the glacier margin ($\overline{V} > 0$).

Climatological flux footprints were calculated for each station and for katabatic and non-katabatic flow periods, using the two-dimensional footprint parametrization of Kljun et al. (2015). Here we used a boundary layer height of 100 m and surface roughness of 0.004 m (Greuell and Smeets, 2001; Fitzpatrick et al., 2019; Nicholson and Stiperski, submitted). We use this

model as a first guess for the flux source area only, given a number of uncertainties. First, the model was not specifically designed for use in sloping terrain, second, our dataset is not allowing us a reliable estimate of the boundary layer height, and third, estimation of surface roughness for katabatic flow is challenging. Indeed, a lower surface roughness would cause



footprint area to increase. Sensitivity analysis, however, shows that this increase is no considerable even when reducing the surface roughness by an order of magnitude.

As we are interested in the interplay of katabatic flow with other local circulation patterns, in this study we focus on five days in August 2018 that meet three criteria: (1) good data quality at all the stations, (2) predominantly clear sky conditions and (3) flow is characterized by a significant shift of wind direction from katabatic down-glacier flow direction to a westerly or northwesterly flow during the day.

Pure katabatic flows are defined as time periods with persistent flow direction from southwest (approximately 200° at station
TT3). Disturbed situations are defined by a deviation of wind direction of more than 50° from the dominant katabatic flow direction during a time period longer than 30 minutes. Following these criteria, the analysis of turbulence data was performed for the following five days:  4, 5, 11, 15 and 20 August (referred to as day 1-5). During these days, persistent katabatic flow was disturbed by westerly winds or up-valley flows (strong shift of the dominant wind direction during the day from southwest to the westerly or easterly wind sector). We thus distinguish between katabatic and disturbed situations (distinct deviation of
the dominant wind direction from the katabatic flow direction). Note that the upper turbulence sensor (CSAT, level 2) at TT2 was not working until 7 August, due to a faulty cable which had to be replaced. During this period turbulence profiles were analyzed for stations TT1 and TT3.

Horizontal heat advection was calculated between transect stations TT1 and TT2 (distance of 65 m) and TT2 and TT3 (distance 114 m). We only calculated heat advection at the lowest level above ground as air temperature was measured only at this height
(see Fig. 1c). Heat advection $HA$ was calculated as passive advection of temperature $T$ $(y, t)$ carried along by the mean y flow component $\bar{V}$ using finite differences: $HA = -\frac{\Delta T}{\Delta y}\bar{V}$

Here the relevant mean wind speed is calculated as the mathematical average of the $y$ wind component between the pairs of stations.

Similarly, the vertical flux divergence $FD$ of the vertical sensible heat flux ($\overline{w'T'}$)  was calculated between the two
measurement levels as:

$$FD = \frac{\Delta \overline{w'T'}}{\Delta z}$$

According to Denby (1999) and Grachev et al. (2016) profiles of streamwise momentum ($\overline{u'w'}$) and streamwise heat ($\overline{u'T'}$) flux provide an approximation of the vertical location of the jet height because typical turbulence profiles observed in the presence of low-level jets show a change in sign of  the streamwise momentum flux (negative below and positive above) and heat flux
(positive below and negative above) at the wind speed maximum. Turbulent kinetic energy reaches its local minimum at jet height while the temperature variations reach a local maximum. Following these observations, the position of the jet-speed maximum can be estimated by linear interpolation between two heights where momentum fluxes are measured (Grachev et al., 2016). This estimate assumes that the momentum flux decreases linearly, and can be applied confidently only if the jet maximum height happens to be between the two measurement levels. We use this indirect estimate of jet maximum height





from the turbulence profiles at the across-glacier transect to examine the change of katabatic flow depth across the glacier and its disturbance by heat advection from the glacier surroundings. In this case the fluxes are not rotated into the new coordinate system but are streamwise.

## 3 Results

### 3.1 Mean flow characteristics across the glacier

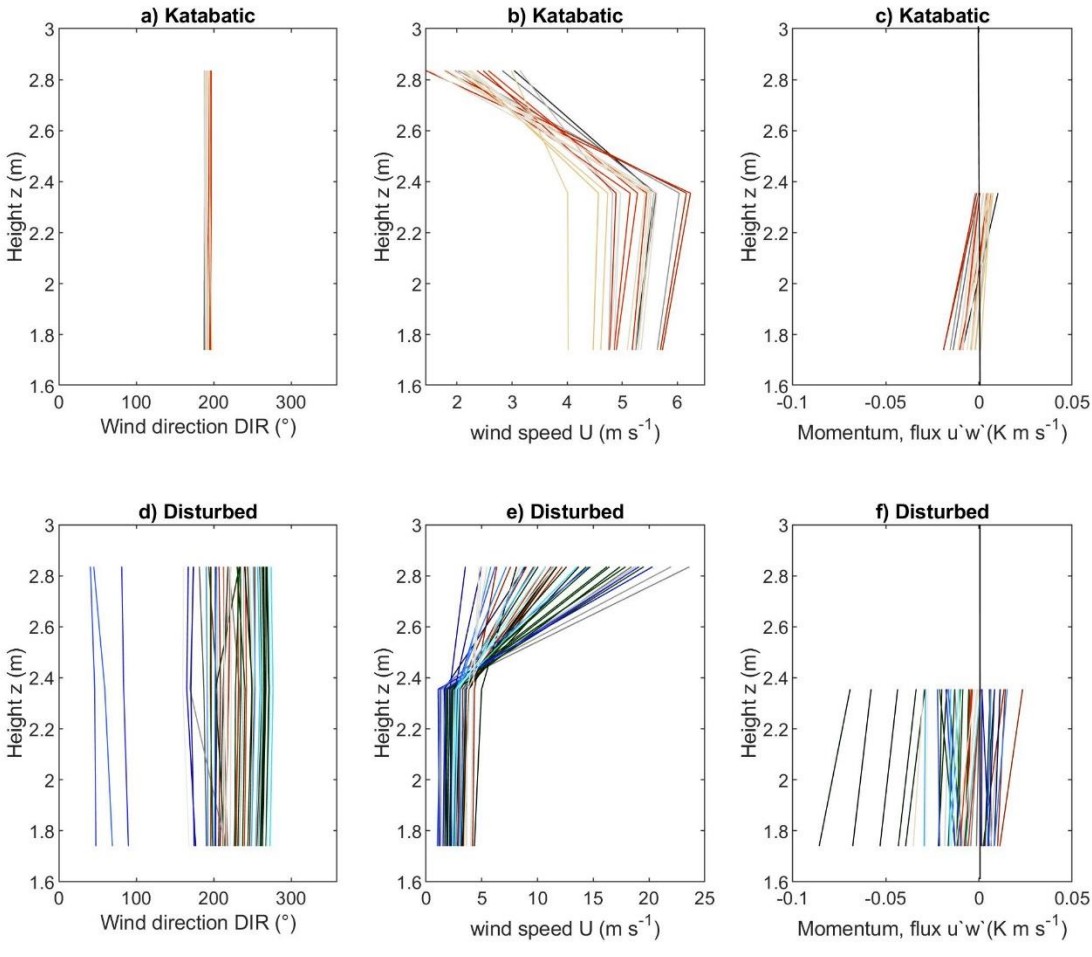


*Figure 2: 30 minutes averaged profiles of wind direction (a, d) and wind speed (b, e) for katabatic (a-c) and disturbed (d-f) situations during five days in August 2018 obtained from the mobile wind tower and station TT3. Streamwise momentum fluxes are shown for two measurement levels obtained from station TT3 (c, f). Note that data was only considered as pure katabatic*



*if data showed katabatic flow during the entire 30 minutes periods. Colors indicate different measurement days (grey=day1,*
*red=day 2; green=day3; blue=day 4; brown=day5).*

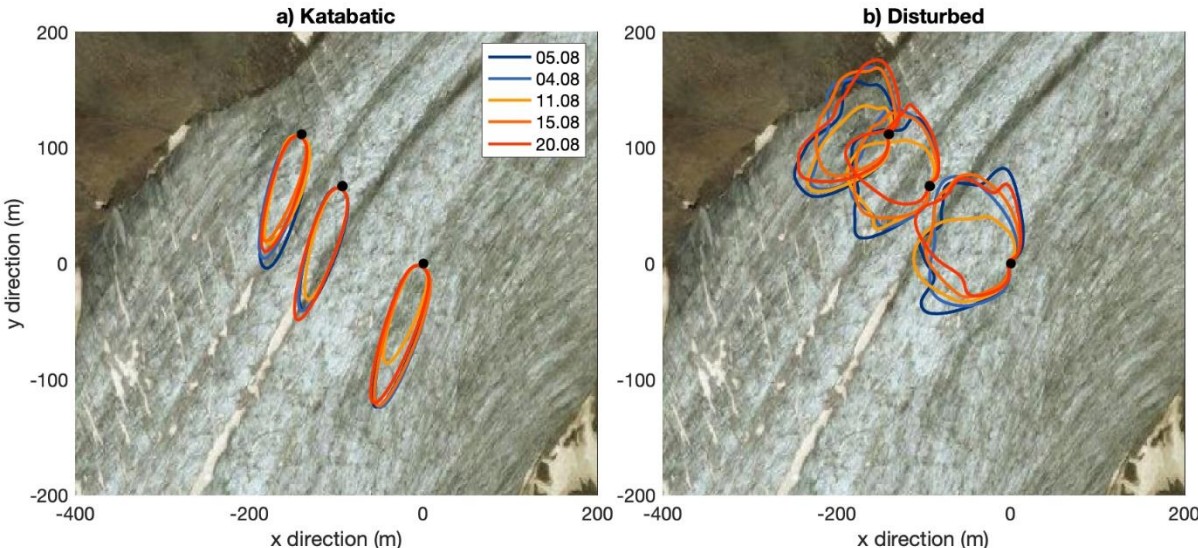

*Figure 3: Climatological flux footprints for transect stations TT1-TT3 and for a) katabatic and b) disturbed conditions.*
*Background images © Microsoft BingTM Maps Platform Arial screen shot(s) reprinted with permission from Microsoft*
*Corporation.*

Profiles of wind speed, wind direction and streamwise momentum flux measured at TT3 is shown in Fig. 2. Furthermore, climatological flux footprints for all three transect stations are presented in Fig. 3 describing the upwind area where 90% of measured fluxes measured at level 1 are generated. During periods defined as pure katabatic flow, wind directions are quasi
constant, varying only between 190° and 200° at TT3 (Fig. 2a), while wind direction is much more variable during periods of disruption of the katabatic flow (Fig. 2d). Highly consistent flux footprints during katabatic flows (Fig. 3a) further highlight the consistent wind direction during katabatic flows. Footprints vary between a few tens of meters to approximately 100 m and are largest at the centerline. During disturbed situations footprints show a dominance of westerly to northwesterly flows but with a high temporal variability at all stations.
Wind speed profiles differ substantially between pure katabatic and disturbed situations (Fig. 2 b, e). For katabatic situations wind speed profiles indicate a distinct low-level wind speed maximum within the lowest 2.9 m above the surface (Fig. 2b). The shape of wind speed profiles suggests low-level jets between 1.7 and 2.3 m, with observed wind speed maxima between 4 and 6 m/s. In contrast, profiles during disturbed situations show small wind speeds within the lowest 2.3 m above ground compared to a strong increase in wind speed at the level 3 (Fig. 2e), reaching maxima of up to 20 m/s. In the near surface layer,


the wind speed gradients are small, and wind speed increases with height with no evidence of low-level jets within the height range of our measurements.

Wind speed profiles measured at all stations during katabatic and disturbed situations are shown in Fig. 4 in order to show how the pattern of katabatic and disturbed airflow evidenced at TT3 compares across the other glacier observation sites. During katabatic conditions, highest peak wind speeds were observed along the centerline at stations TT3 and WT1 (Fig. 4 c, e). Those

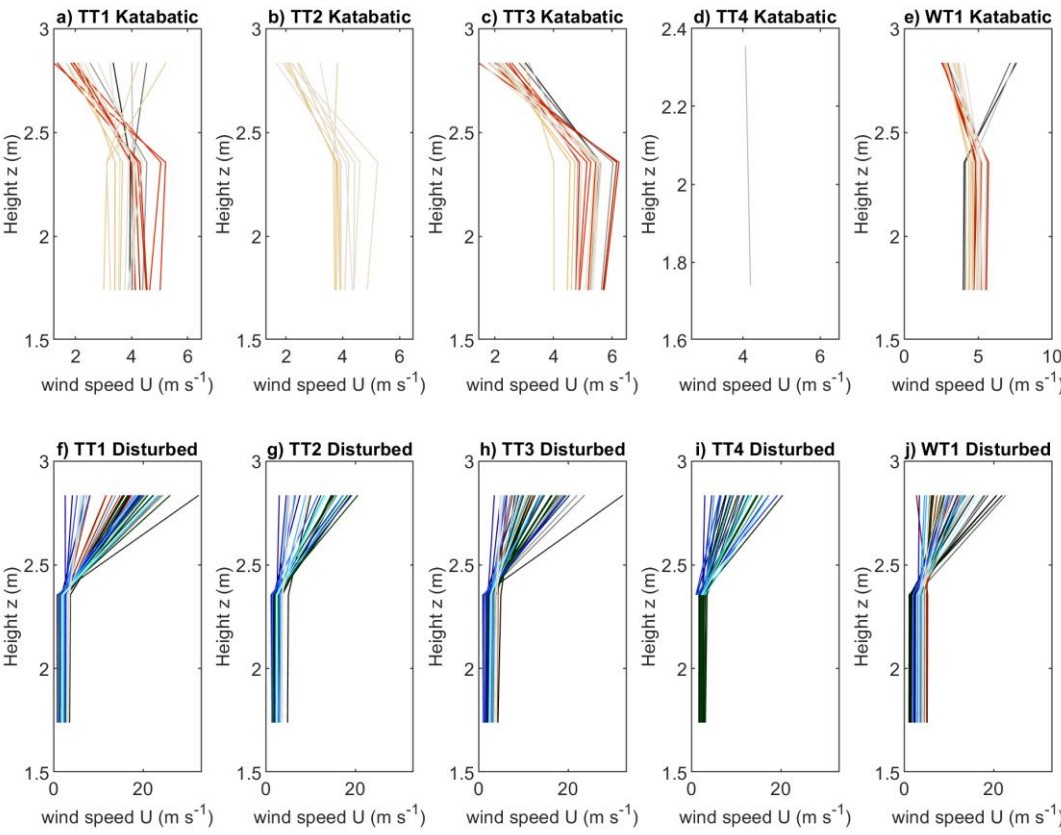


*Figure 4: 30 minutes averaged wind speed (U) profiles for katabatic (a-e) and disturbed situations (f-j) for all stations (TT1, TT2, TT3, TT4, WT1) during five days in August 2018. Note that only data was considered as pure katabatic if data showed katabatic flow during the entire 30 minutes periods Colors indicate different measurement days (grey=day1, red=day 2; green=day3; blue=day 4; brown=day5).*


persistent katabatic flows at the centerline are also indicated by largest footprints at TT3 and decreasing footprints towards the glacier margin. Wind speed profile characteristics are typically similar for all stations at the along- and across-glacier transects (Fig. 4 a-e), although there are some situations when the stations at the glacier margins WT1 and TT1 (Fig. 4a, e), do not show a significant decrease in wind speed at level 3 or even showed an increase in wind speed at this level. This different behavior





might be explained by disturbances from the non-glacierized surrounding at these two stations. Indeed, wind speed at these marginal stations tends to show more variability, especially at level 3, than in the more centrally located stations.

During disturbed situations (Fig. 2d-f; Fig. 4f-j), wind profiles show at all sites a much stronger temporal variability of wind direction which is also indicated by strong variation of the footprint (Fig. 3 b). Flux footprints tend to be smaller during disturbed situations. The footprint is reaching off the glacier area only for TT1, while for the other stations the footprint of turbulence measurements is limited to the glacierized area. The low wind speeds and wind speed gradients below 2.3 m above ground, and strong increase in wind speed at level 3 (difference of up to 15 m/s for 0.5 m vertical distance) is evident at all stations across and along the glacier (Fig. 4 f -j) suggesting the presence of a secondary larger-scale wind system. Unfortunately, no turbulence sensors were installed at heights above 2.3 m, thus we have no information on the turbulence characteristics at these heights which would provide more information on the origin of the elevated flow. Based on the very strong upper-level winds and the predominantly measured westerly to northwesterly wind direction we assume that these westerly flows were connected to a large-scale westerly circulation that developed over the day and disturbed the katabatic flow (Whiteman and Doran, 1993). A second explanation, less likely due to the very high wind velocities at the highest measurement level, is the development of cross-valley circulations caused by the curvature of the valley (cf. Weigel and Rotach 2004) at the lower parts of the glacier or a thermal flow originating from the Langtalerjochferner. This extreme increase of wind speed with height is confirmed by preliminary numerical simulations (not shown).

### 3.2 Turbulence profiles and flux footprints of pure katabatic flows and disturbed situations across the glacier

Vertical profiles of streamwise momentum fluxes are shown for station TT3 (glacier centerline) in Fig. 2 c, f. and for TT1 and TT3 in Fig. 5 to demonstrate spatial and temporal patterns along the across-glacier transect. Colored lines represent 30-minute averages of turbulence data. As not all three transect stations were properly working during the five days of interest we present data from stations TT1 and TT3 in Fig. 5 to compare air flow and jet height evidence along the across-glacier transect.

During the katabatic flow situations, streamwise momentum fluxes measured at the centerline stations (TT3) clearly changed from a negative (downward) to a positive flux (upward) between the lower and the upper sensor (Fig. 2c). This behavior is consistent with idealized turbulent mixing and is a strong indication of the presence of a wind speed maximum between the two measurement heights of 1.7 and 2.3 m above the ice surface. The height of minimum fluxes, indicating the jet height of the katabatic flow can be crudely detected by linearly interpolating between these measurement points. The magnitude of momentum fluxes at the two measurement levels is smaller during katabatic situations than observed during disturbed situations (Fig. 2 f), as fluxes typically reach their minimum close to the jet height. On the contrary, during disturbed situations, strong wind speed gradients are observed above the turbulence measurements at TT3 explaining the more frequently observed higher negative momentum fluxes without an observed change in sign with increasing height (Fig. 2e; Fig. 5h). Indeed, no pronounced wind speed maximum was observed at turbulence measurement heights during disturbed situations. Some of the observations during disturbed situations, however, even show positive momentum fluxes at both measurement levels of turbulence (1.7 and 2.3 m), particularly at TT1, closest to the glacier margin (Fig. 5d). These positive momentum fluxes are a





sign for the presence of a local layer with decreasing wind speed below 2.3 m above the ice surface indicating a local very shallow katabatic layer during some of the disturbed situations.

At the central station momentum fluxes during katabatic flow clearly changed from negative to a positive flux between the lower and the upper sensor suggesting a jet height between the two measurement levels. Momentum fluxes at station TT1 show more frequently positive momentum fluxes at both measurement levels indicating that measurements were conducted above a primary low-level jet height. Well-developed katabatic flows at the centerline also showed higher wind speeds and higher streamwise momentum fluxes particularly at the lower measurement level. In addition, Jet heights are found to be more

consistent at TT3 with most of the profiles indicating a jet height between level 1 and level 2 (approx.1.7 – 2.3 m). At TT1 where strong temporal variability of momentum flux profiles indicates jet heights lower and higher than level 1 and 2. Profiles of streamwise momentum fluxes at the centerline show a steeper gradient of momentum flux in the layer below the wind-speed maximum than observed at the margin station where the lower measurement level was predominantly located approximately at the jet height. Measurements thus show that, as expected, turbulent fluxes changed more slowly with height (i.e. small flux

divergence) in the region above or at the local wind speed maximum than below it.

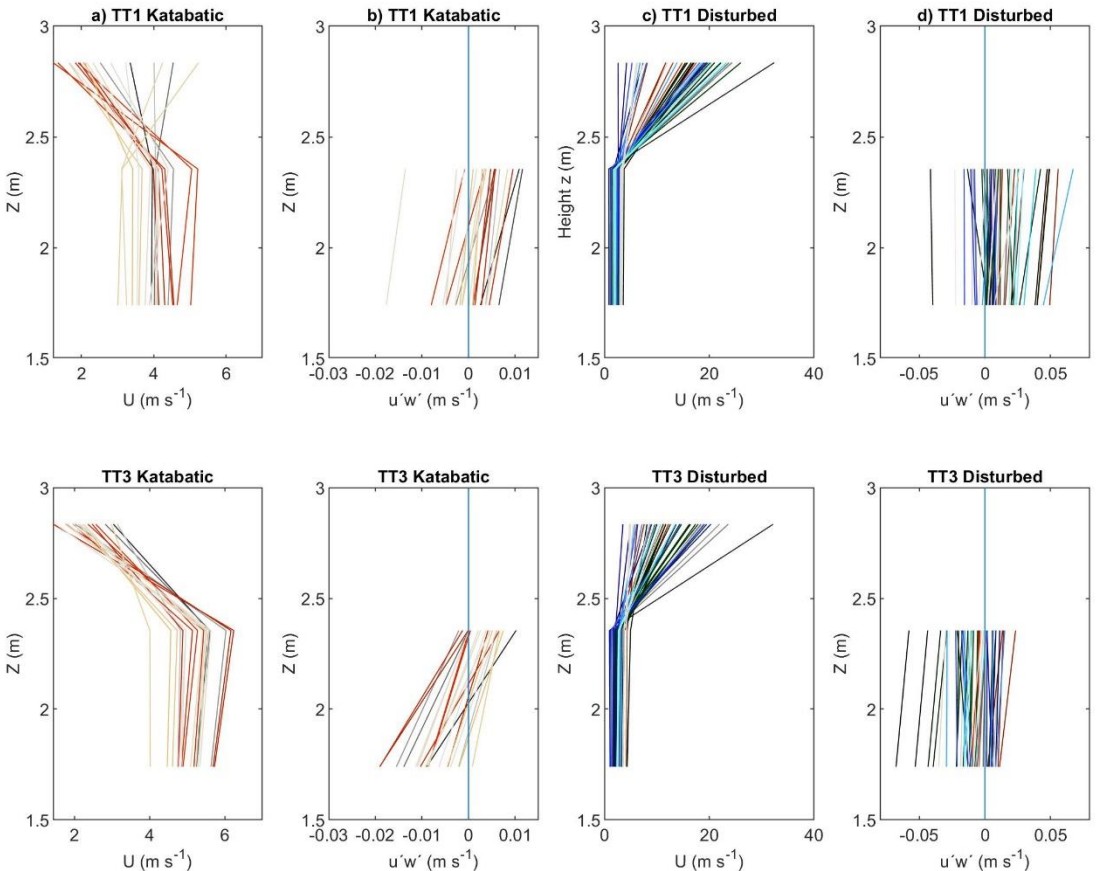

*Figure 5: Profiles of 30 minutes averages of wind speed (U) streamwise momentum flux ($\overline{u'w'}$) measured during pure katabatic flow and during disturbed flow conditions on 2018-08-20 at transect stations TT1 and TT3. Note that only data was considered as pure katabatic if data showed katabatic flow during the entire 30 minutes periods. Colors indicate different measurement*

*days (grey=day1, red=day 2; green=day3; blue=day 4; brown=day5).*

There are considerable differences in the turbulence characteristics observed for katabatic flows and disturbed situations. First, momentum fluxes were much higher for the disturbed situations indicating a significantly stronger turbulence and transport of momentum. Second, momentum fluxes do not frequently change sign between the two measurement levels. In combination

with the small vertical flux divergence between the two measurement levels turbulence data during disturbed situations indicate that measurements at these heights were conducted within a statically stable layer not much affected by a katabatic jet. Finally, the temporal variability of flux profiles increased significantly for disturbed situations which might be a result of a stronger intermittency of the flow during disturbed situations in which high momentum air is mixed into the stable katabatic layer. We





also observed similarities in the turbulence structure between the two different situations. Similar to katabatic situations,
momentum fluxes are predominantly negative at the centerline, but were fluctuating between negative and positive directions
at the margin station. The strong temporal variations of the sign of the momentum flux at the margin station suggest the
presence of an intermittent flow with a windspeed maxima below the turbulence measurement levels for specific time periods.
No measurements of wind speed profiles at high enough resolution close to the ground are available, however, to test this
hypothesis for the station close to the glacier margin.

## 3.3 Evolution of air temperature and heat exchange connected to prevailing wind conditions

### 3.3.1 Mean air temperature and relative humidity

The focus of this section is on the change of the local thermodynamic characteristics at the glacier driven by local flow
conditions. Figure 6 presents near-surface normalized air temperatures and wind directions measured at stations at the across-
glacier transect (TT1, TT2, TT3) and the along-glacier transect (TT4, TT3, WT1). In order to allow a comparison between air
temperature evolution during different days air temperatures were normalized by mean daytime air temperature of the
respective day and station measured between 10 AM – 6 PM. The color codes indicate the deviation of the measured wind
direction at TT3 (Fig. 6a), TT4 (Fig. 6b) and WT1 (Fig. 6c) from the dominant katabatic flow direction (defined 200° at the
across-glacier transect). Strongest deviation from the dominant katabatic flow direction (blue colors) showing northerly to
north-easterly flows indicate the presence of up-valley flows (red colors). Deviations of approximately 70° correspond to a
westerly flow which appears as an across-glacier flow (light blue colors).

During katabatic situations normalized air temperatures stayed at low values with smaller air temperatures along the centerline
of the glacier (TT3, TT4) than at the margin station TT1. Both stations located approximately at the centerline of the glacier
(TT3, TT4) featured highest wind velocities driving a rather homogeneous temperature distribution along the glacier centerline.
The spatial variability of air temperatures during katabatic situations were smaller between stations located at the along- than
at the across-glacier transect (Fig. 6 a, b) despite larger differences in altitude for the along-glacier transect.

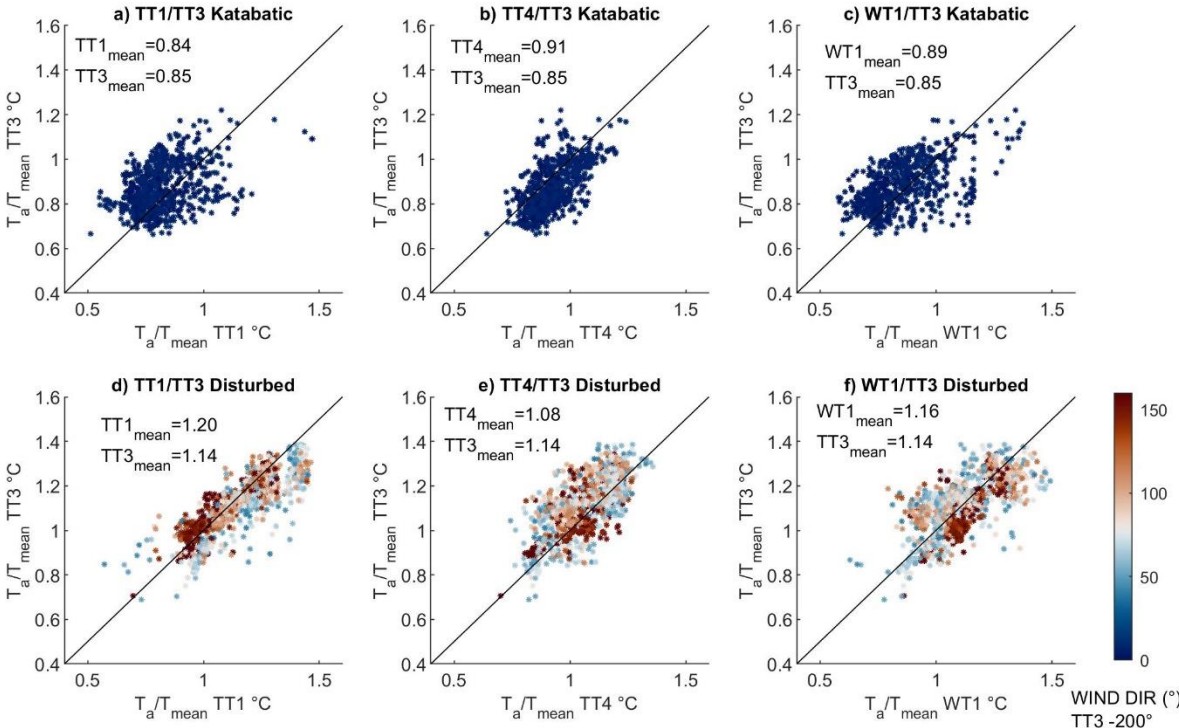

*Figure 6: Mean air temperatures normalized by mean daytime air temperature at the respective stations TT1, TT3, TT4 and WT1 are shown for five selected days with a clear shift in wind direction. Comparison of stations TT1 and TT3 (a d), TT4 and TT3 (b, e) and WT1 and TT3 (c, f) are shown for katabatic (a-c) and disturbed situations (d-f). Color codes indicate deviation from dominant katabatic flow direction (defined as 200°) at TT3 in degrees for all of these five days.*

As soon as the katabatic flow was disturbed by the westerly wind, local wind directions became much more variable ranging between west-northwest and northeast (deviations from katabatic wind direction ranging 50° – 180°). The change in wind directions evidenced by all across-glacier transect stations coincided with a significant increase in the near-surface air temperature of several degrees (Fig. 6d-f) and a decrease in relative humidity of 9 to 13 % on average (Table 1). The change in air temperatures showed strong spatial differences, with strongest air temperature rise in the peripheral areas (TT1, Fig. 6a, d) and significantly smaller temperature rise along the glacier centerline (TT3, TT4; Fig. 6b, e). Similarly, the drying out of the near-surface air is stronger in the peripheral zone than at the centerline. Local air temperatures at the higher altitude station TT4 showed the lowest sensitivity to changes in wind direction at TT3. The katabatic flow seemed to persist at the higher-altitude station TT4 when at the same time all transect stations already evidenced a westerly flow (Fig. 6b). Data thus suggest that the station TT4 was more sheltered from westerly flows than stations located at lower parts of the glacier. Air temperatures at the glacier tongue (WT1) appeared to be strongly affected by up-valley flows (Fig. 6f). Measurements thus reveal a higher





impact of near-surface air warming during westerly flows on stations located in areas close to the glacier margin such as in the
peripheral areas (TT1) and at the glacier tongue (WT1) (Fig. 6a, b).

Note that the wind system often changed between katabatic and disturbed flows within short time periods of a few minutes. During these intermittent situations, short-term southwesterly flows (defined as katabatic flow direction) showed higher air temperatures than typically observed during persistent katabatic flow situations (normalized air temperatures above 1), which were most probably still influenced by the disturbed flow. This might partly explain the scatter of air temperatures for the
katabatic flow.

These spatial differences in flow development might explain a larger spatial variability of the air temperature field along the centerline of the glacier during prevailing westerly flows than during katabatic flows. The strong sensitivity of the mean air temperature to the presence or the disturbance of an along glacier katabatic wind indicates that well-developed katabatic winds decouple the local near-surface air temperature at the glacier from the warmer surrounding air. This is well reflected by
significantly lower air temperatures during well-developed katabatic flows. Measurements also suggest that the local disturbance by the across-glacier flows promote the advection of warm air towards the glacier with strongest effects at the peripheral zones of the glacier.

*Table 1: Averaged values of normalized air temperatures and wind velocities and of turbulent vertical heat flux (w'T') at*
*stations TT1 and TT3. Correlation coefficients between 1) vertical turbulent heat flux and normalized air temperature, 2) between vertical turbulent heat flux and normalized wind velocity, 3) vertical turbulent heat flux and horizontal heat advection and 4) vertical turbulent heat flux and wind velocity component along the transect. Values are provided for Katabatic (K) and disturbed (D) situations.*

| | Mean | | | | | | | | Correlation Coefficients | | | | | |
|---|---|---|---|---|---|---|---|---|---|---|---|---|---|---|
| | $U/U_{mean}$ | | $T/T_{mean}$ | | RH (%) | | w'T' | | w'T' , $T/T_{mean}$ | | w'T', $U/U_{mean}$ | | HA, w'T' | w'T', $U_T$ |
| Situations | K | D | K | D | K | D | K | D | K | D | K | D | D | D |
| **TT1** | 1.39 | 0.84 | 0.84 | 1.10 | 80 | 67 | -0.035 | -0.041 | -0.21 | -0.2 | -0.21 | -0.42 | 0.19 | 0.17 |
| **TT3** | 1.34 | 0.85 | 0.88 | 1.14 | 79 | 70 | -0.037 | -0.051 | 0.06 | 0.12 | -0.26 | -0.47 | 0.31 | 0.56 |






### 3.3.2 Vertical heat exchange

We analyzed turbulent sensible heat fluxes at all four turbulence stations installed at the across-glacier transect (TT1-TT3) in order to address how increasing air temperatures during disturbed situations affect local heat exchange processes, potentially promoting ice melt. In glaciology it is conventional to give heat fluxes in terms of gains and losses with respect to the glacier surface, such that a downward flux, termed negative in atmospheric science is given as a positive flux in glaciology as it represents an energy contribution to the glacier surface. We are following the convention of atmospheric science, where a negative sensible heat flux indicates a flux directed towards the glacier surface. As most turbulent flux parameterizations assume a linear relationship between turbulent fluxes and wind speed, we plotted turbulent sensible heat fluxes against normalized wind velocity measured at stations TT1 – TT3 in Fig. 7. The color of each data point provides additional information on the normalized air temperature for katabatic situations (a-c) and disturbed situations (d-f).

Turbulence data reveal higher vertical turbulent sensible heat fluxes during disturbed than during katabatic situations. Higher heat fluxes coincide with higher air temperatures particularly at the margin station (Fig. 7 d-f), also reflected by mean turbulent heat fluxes averaged over all five days during disturbed situations (-0.051) being significantly higher than during katabatic situations (-0.037). With the melting surface of the glacier at zero degrees, the increasing near-surface temperature gradients coincided with an increase of downward turbulent heat flux. As already mentioned, near-surface wind speeds during disturbed situations were typically lower than the daytime average wind speed. Sensible heat fluxes, however, show a strong correlation with the low-level wind speed during disturbed situations (R(w'T'- $U/U_{mean}$), Table 1). Most of those katabatic situations coinciding with higher air temperatures also coincided with very low normalized wind velocities. This again indicates that these individual katabatic flow situations with high air temperatures can be rather characterized as intermittent flows than well-developed katabatic flows.

During disturbed situations turbulence data showed small spatial differences of turbulent heat exchange at the across-glacier transect. Fluxes are particularly similar at TT1 and TT3 despite significantly higher air temperatures observed at TT1. While air temperatures were lower at TT3 than at TT1, higher wind velocities at the centerline appeared to promote heat exchange there. This is also confirmed by statistics shown in Table 1. At the central station wind shows higher correlations with turbulent heat fluxes than at the margin station particularly for the katabatic situation. In contrast to the margin station TT1 which shows similar correlations between air temperature and turbulent heat fluxes for both situations, the central station TT3 shows no correlation between air temperatures and vertical heat fluxes.





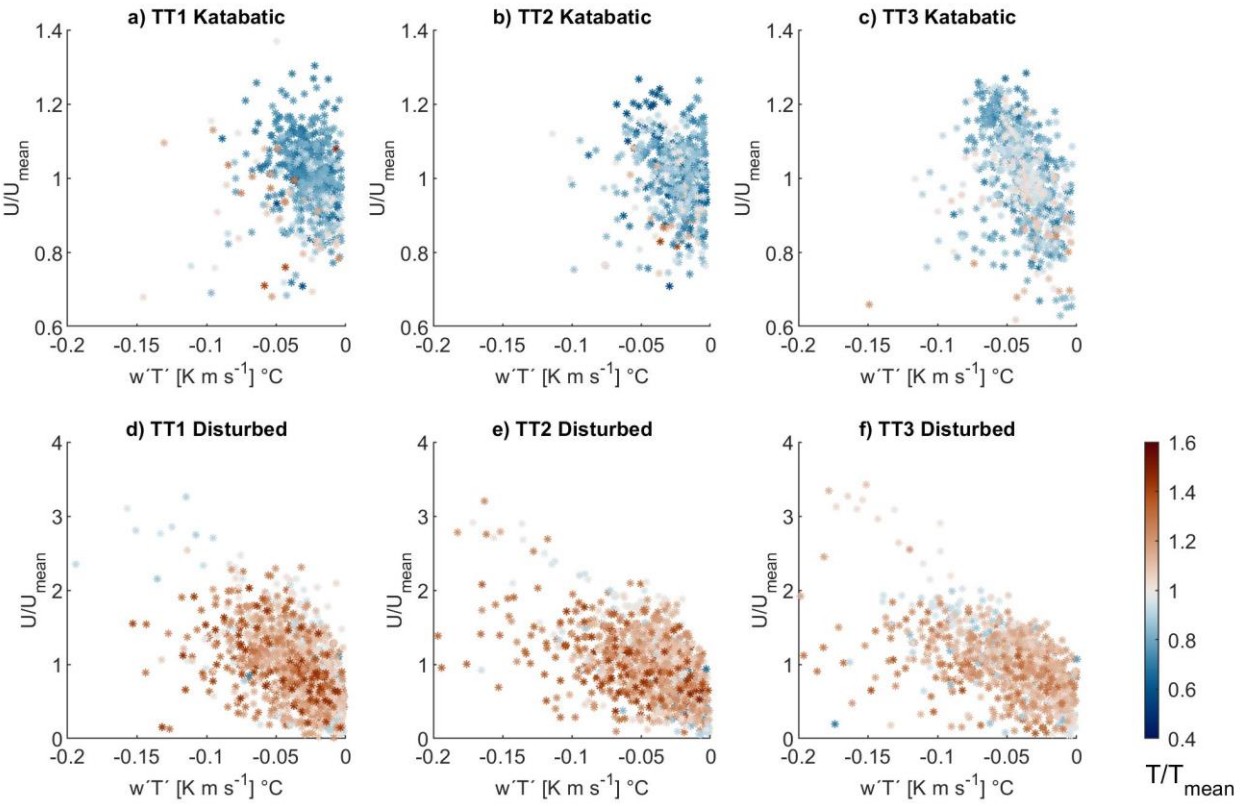


*Figure 7: Vertical heat flux plotted against wind speed normalized by 1-minute averaged wind speeds shown for stations TT1 –TT3 for katabatic situations (a-c) and disturbed situations (d-f). Color codes show air temperature normalized by minimum daytime air temperature.*

### 3.3.3 Lateral heat advection

Measurements of air temperatures suggested a strong influence of warm air advection during larger-scale northwesterly flows disturbing the katabatic flow at the glacier forming an across-glacier flow. In a next step we quantify the horizontal warm air advection HA for across-glacier flow conditions. A transect consisting of three stations was aligned in a northwesterly orientation allowing the calculation of HA between neighboring stations during across-glacier flows. Figure 8 illustrates the advection of heat as a function of the deviation of the flow from the dominant katabatic flow direction. The color of each data

point indicates air temperature differences between neighboring stations TT1-TT2 and TT2-TT3 (Fig. 8 a, b) and mean V-component in the direction of the transect (Fig. 8 c, d). Positive horizontal differences in air temperature result from warmer air temperatures at the margin stations and a decrease towards the centerline. We defined a negative V–component along the transect directing from TT1 to TT3 (Fig. 8 c, d). Thus, a negative advective heat flux indicates the advection of warm air from





the peripheral zones of the glacier towards the glacier centerline (positive air temperature differences and negative V-

component). Positive values of heat advection identify situations when colder air was advected along the TT1-TT3 transect

(negative V-component and negative temperature gradient). Situations with positive V-component along the transect were

excluded from this analysis as these were situations when wind direction was east to southeast. For these situations the transect

was not properly aligned.

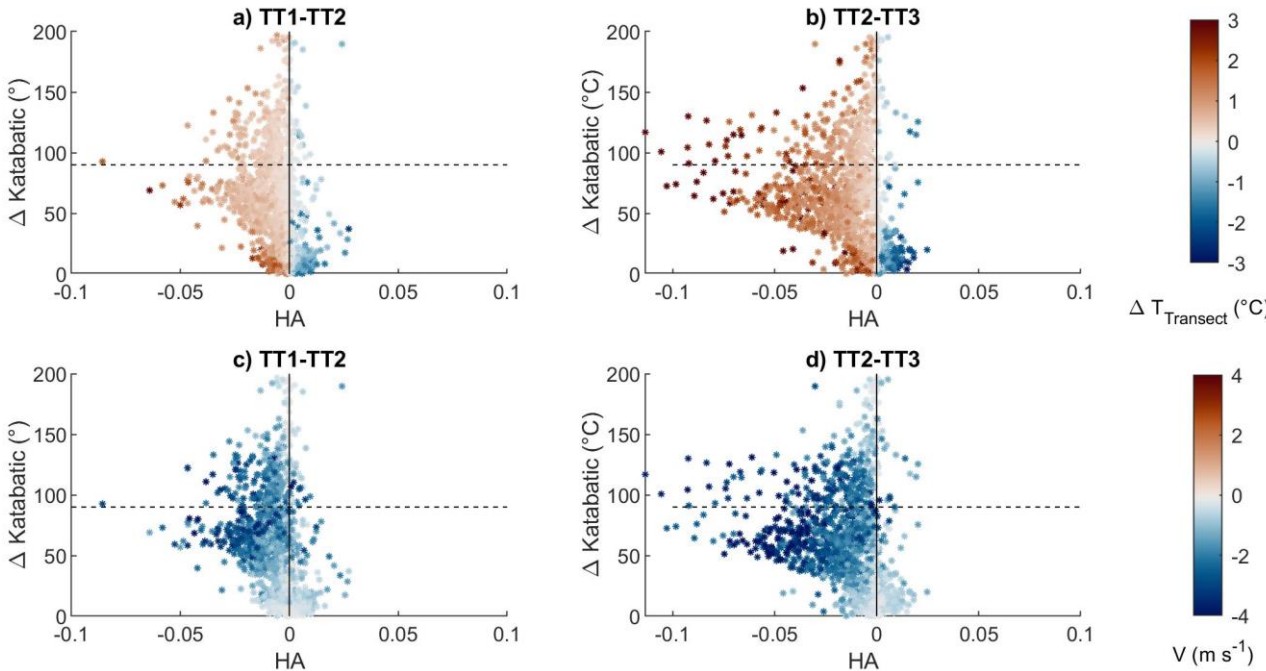


***Figure 8:*** *Horizontal advection of heat (HA) calculated between stations TT1 and TT2 (a) and TT2 and TT3 (b) plotted against*

*deviation from katabatic wind direction for 5 selected days with periods of clear deviation from the dominant katabatic flow*

*direction. Color codes indicate the measured air temperature difference between stations TT1 and TT2 and TT2 and TT3 (a,*

*b) and the wind velocity component along the transect V (wind speed component along the Transect) (c, d). Positive values of*

*air temperature difference indicate higher air temperatures at the station closer to the glacier margin. Negative wind velocity*

*component indicate wind from station TT1 to TT3. The dashed line indicates the deviation of the wind direction 90 degree*

*from the dominant katabatic flow which is the orientation of the transect.*

Strong positive horizontal air temperature gradients along the transects occurred for westerly to northwesterly winds (60 – 90°

deviation from katabatic). Horizontal heat advection HA increased with temperature differences and V-component along the

transect line and increased from the peripheral stations towards the centerline station TT3. Therefore, peak HA at the centerline

can be explained by stronger temperature difference between the middle and the central station (TT2 and TT3) than between



the two more peripheral stations. Furthermore, strongest temperature differences between all stations concurred with peak V-components in the direction of the transect of more than 4 m/s. These V-components increased towards the centerline. On the

contrary, the small V-components at the peripheral station TT1 indicates that the margin stations are more sheltered from the strong large-scale synoptic wind than the station at the centerline. This is in contrast to earlier numerical results of Sauter and Galos (2016) who suggested that well-developed katabatic flows at the centerline of glaciers prevent warm air advection from the surrounding. This conclusion seems not to be valid for synoptic winds strong enough to disturb the katabatic flow along the centerline.

Negative air temperature gradients (colder air temperatures at the peripheral areas, blue colors) were only measured during short time intervals. For some cases, warm and cold air advection even occurred during intermittent flow situations (changing between southwesterly and northwesterly flow directions within short time) but with much smaller wind velocities than observed during well-developed katabatic flow situations. The heat advection during these situations, however, was much weaker.

We are interested in the efficiency of the horizontal heat transport to warm near-surface air layers and thus to indirectly promote turbulent heat exchange towards the ice surface contributing to the surface energy balance. We therefore analyzed the relationship between horizontal heat advection HA (TT1-TT2 and TT2-TT3), the vertical turbulent heat flux and the V-component along the transect (color code), illustrated in Fig. 9. Additionally, correlation coefficient R between those variables are provided (Fig. 9; Table 1). Note that for this analysis we considered only data with evidence of horizontal heat advection

(HA) along the transect (V-component along the transect larger than 1 m/s and positive air temperature differences). Highest correlation between HA and $\overline{w'T'}$ was found at TT3. Thus, at the centerline (TT3) heat advection appears to be most efficient in enhancing turbulent heat exchange towards the glacier surface by enhancing near-surface temperature gradients. Similar to heat advection, peak vertical turbulent heat fluxes coincided with peak V-component at the centerline. Correlation coefficients $R_{(w'T', UT)}$ were high between TT1-TT2 and TT2-TT3 station pairs with a slightly higher value for stations closer to the

centerline. Turbulent heat fluxes showed slightly smaller values at TT1, coinciding with significantly smaller wind speeds. Furthermore, the correlation between wind speed and vertical turbulent heat flux at the peripheral station was very small highlighting the small potential contribution of wind on the downward turbulent heat flux at TT1. In the absence of strong winds at the margin areas, the higher near-surface air temperatures induced by heat advection from the ice-free surrounding of the glacier appear to drive the vertical turbulent heat flux at the peripheral zone of the glacier by increasing near-surface vertical

temperature gradients. In contrast, at the centerline strong winds not only promote heat advection but also promote maximum downward turbulent heat exchange reflected by higher correlation coefficients. Consequently, at the glacier centerline (TT3) stronger winds enhance both the heat advection and the turbulent heat exchange. Near-surface air at the centerline is thus cooled by strong downward sensible heat fluxes which might contribute to cooler air temperatures there.





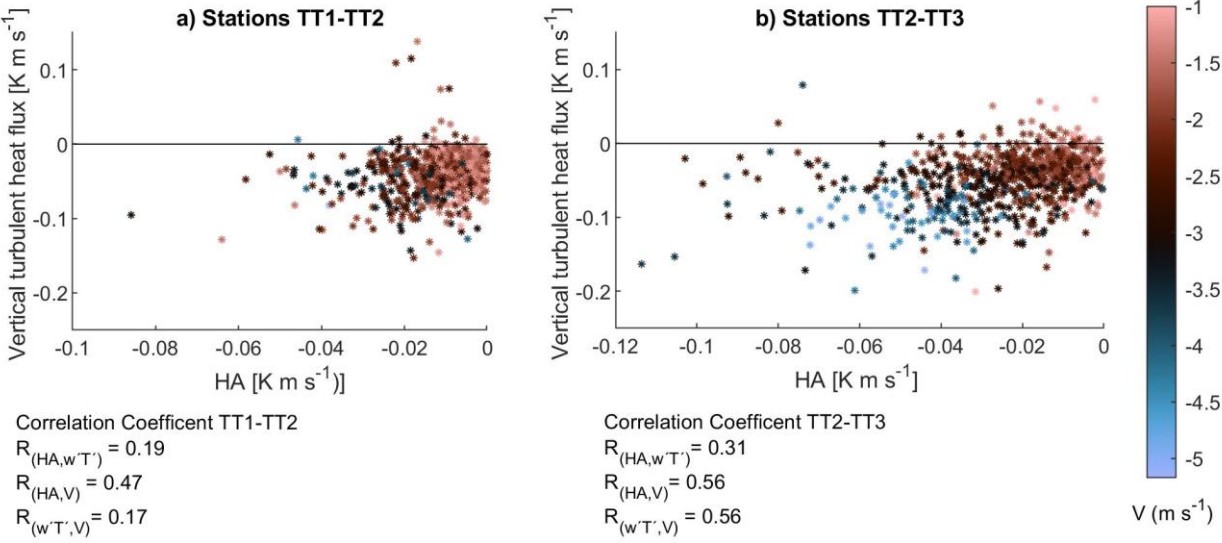

***Figure 9:*** *Horizontal advection of heat (HA) between stations 1 and 2 (a) and 2 and 3 (b) plotted against turbulent vertical heat flux for 5 selected days with clear shift in wind direction. Color codes show the measured wind speed component along the transect. Negative values present wind direction from the margin towards the centerline (northwesterly winds). Correlation coefficients R are provided between the different variables HA, $\overline{w'T'}$ and V (wind speed component along the Transect).*

## 4 Discussion

In the presence of katabatic winds, similarity-based scaling parameterizations used to link the surface energy balance to the flow or the estimation of surface turbulent fluxes from turbulence measurements are not valid (Nadeau et al., 2013; Oldroyd et al., 2014; Grachev et al., 2016). This is because the jet height imposes a strong control on the turbulent structure of the katabatic flow (e.g., Denby and Smeets, 2000; Stiperski et al., 2019a) so that turbulent fluxes in katabatic flows vary strongly with height as a function of the jet height location. Therefore, an estimation of the contribution of turbulent fluxes to the energy balance at the glacier surface is challenging and inferring turbulent surface fluxes from measured fluxes at a certain height will lead to strongly biased surface energy balance calculations. Analysis of momentum flux profiles during katabatic and disturbed situations showed that in the presence of a low-level wind speed maximum turbulent fluxes typically have their local minimum at the jet height and increase below the jet height in line with strong vertical gradients there (Fig. 4). Thus, the magnitude of measured turbulent fluxes strongly depends on the measurement location relative to the jet height. A more detailed analysis on the existence of a jet height during disturbed situations is needed to assess the effect of heat advection during prevailing westerly flows on the heat exchange towards the glacier surface. Figure 10 shows the vertical momentum flux as a function of the sensible heat flux divergence FD and sensible heat flux (color code) at the across-glacier transect stations TT1, TT2 and TT3 for katabatic and disturbed situations. Flux divergence was calculated between the two measurement levels. Positive





momentum fluxes are a sign of decreasing wind speed with height, suggesting the presence of a local wind speed maximum below the respective measurement height. On the contrary, negative momentum fluxes suggest that measurements were performed in a layer with increasing wind speed with height (see also Fig. 5). Katabatic flows typically coincide with strong vertical flux divergences due to strong gradients in wind velocity and air temperature. While these high flux divergences are typically observed in layers where wind speeds strongly increase with height, very small vertical divergences might indicate

either a constant flux layer in the absence of a low-level jet (negative momentum flux), that measurements are conducted close to or above the wind speed maximum (small or positive momentum fluxes) or that strong stability is responsible for strong turbulence suppression (Stiperski et al. 2019a).

We are not only interested in changes in the turbulent structure when changing from katabatic to disturbed situations but also on the effect of heat advection on the turbulent heat fluxes. Turbulence data of katabatic and disturbed situations reveal some

similarities along the transect stations but also strong differences between the different flow conditions (Fig. 10). First, the three transect stations show a similar trend for both situations with an increase of the turbulent heat flux and heat flux divergence from the margin station towards the central station. Second, the transect stations reveal a trend for both situations from more frequently measured positive and small momentum fluxes at the margin to larger and more frequently measured negative momentum fluxes at the central station. On the other side, the largest differences are the much higher magnitudes of

turbulent fluxes of momentum and heat as well as higher flux divergence of turbulent heat fluxes during disturbed situations. In order to assess the effect of heat advection on the heat exchange processes during disturbed situations we focus our analysis on flow characteristics during those conditions (Fig. 10 d-f). During westerly flow situations turbulence data at the centerline of the glacier (TT3) show a strong increase of downward vertical sensible heat fluxes with increasing downward momentum fluxes (negative values) (Fig. 10c). The strongest turbulent fluxes of heat coincided with peak vertical heat divergence. At the

more wind-exposed centerline, negative momentum fluxes and the strong flux divergence indicate that no pronounced katabatic jet is present below the lowest measurement level and that measurements were conducted within a stable atmospheric layer with increasing wind velocities with height featuring strong flux gradients close to the surface. Strong turbulent momentum and sensible heat fluxes combined with strong flux divergence at TT3 suggest very efficient turbulence transfer towards the surface in case of advective situations.

Contrary to the centerline, momentum fluxes measured at the more peripheral stations TT2 and TT1 show a trend towards a higher frequency of positive momentum fluxes with decreasing distance to the glacier margin (Fig. 10 d, e). While the mid-transect station TT2 evidences predominantly negative momentum fluxes with a considerably smaller flux divergence and smaller turbulent heat fluxes than observed at the centerline (Fig. 10 e), the peripheral station TT1 predominantly show positive momentum fluxes suggesting that the lower measurement level was already located above a low-level jet or close to the jet

height which typically features a local flux minimum and small flux gradient. These positive momentum fluxes measured at TT1 coincided with smaller peak turbulent heat fluxes and heat flux divergence than measured at TT3 at the same time. This supports conclusions of Grachev et al. (2016) that turbulent fluxes in the layer below the wind-speed maximum vary with height more rapidly than in the layer above the katabatic jet.

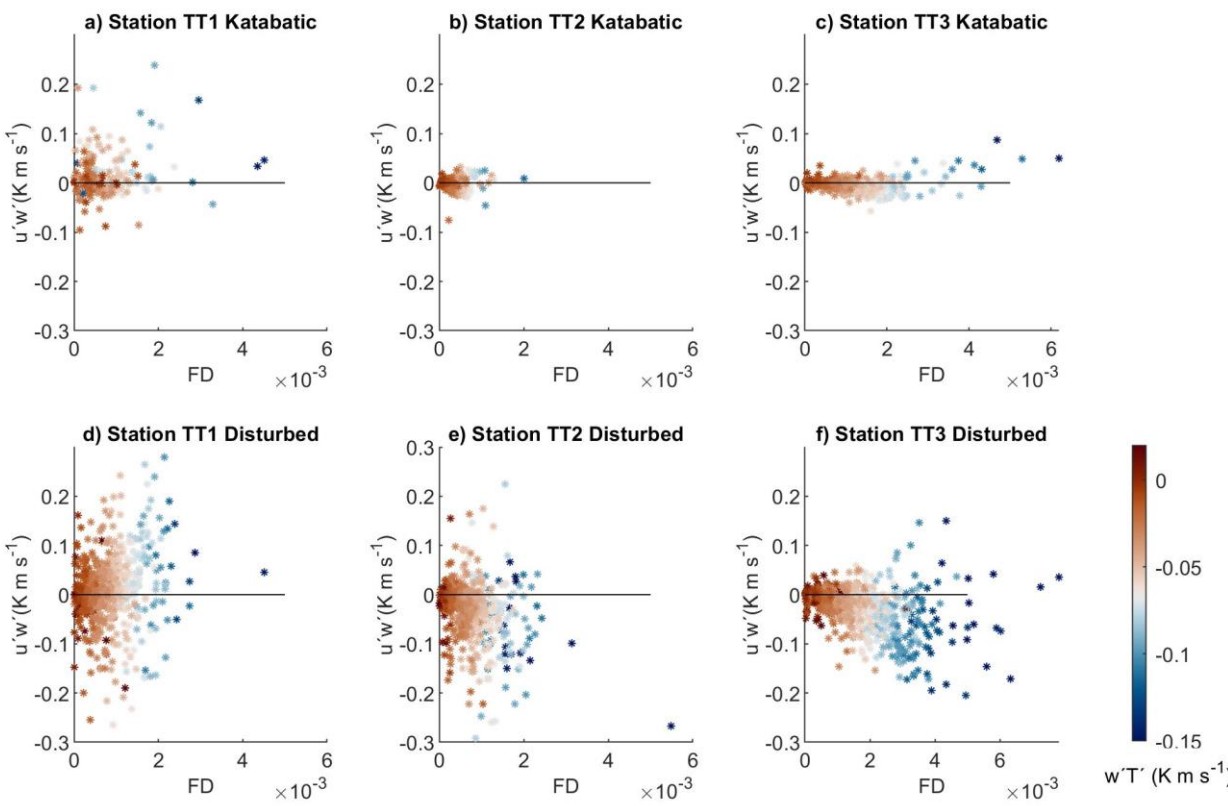


***Figure 10:*** *Vertical flux divergence plotted against vertical momentum flux for stations TT1, TT2, TT3 for katabatic (a-c) and disturbed situations (d-f). Color codes indicate the vertical turbulent heat flux at the corresponding station. Negative values indicate a downward turbulent heat flux. Note that for this analysis we considered only data with evidence of horizontal heat advection along the transect (U-component along the transect larger than 1 m/s and positive air temperature differences).*


The more frequently measured positive momentum fluxes at TT1 and strongly negative momentum fluxes at TT2 and TT3 suggest that the flow at the centerline is more developed than the flow at the margin. Also, lower-level measurements at TT3 revealed significantly higher fluxes than at the peripheral stations where measurements are supposed to be conducted above a very shallow low-level jet. Therefore, the strong increase of the wind speed component towards the centerline (Fig. 8 c, d) and

the potential formation of a very low-level jet height at the margin stations (TT1) suggest strong differences between the flow development at the centerline and in the peripheral zone of the glacier. One possible explanation for the occurrence of the low-level jet at TT1 is the formation of a shallow stable internal boundary layer (SIBL) at the peripheral areas of the glacier when the warm air crosses the peripheral area of the glacier induced by the step of surface characteristics between ice-free surrounding and the glacier (Mott et al., 2015). SIBLs favor the formation of very low-level jets (Mott et al., 2015) as the high





static stabilities of SIBLs over ice are associated with reduced wind velocities near the ground. Above the shallow SIBL the flow field is characteristic of the upstream conditions despite the detachment of the larger-scale flow from the snow surface and its displacement to higher atmospheric levels. An alternative explanation might be that the stronger sheltering of the peripheral areas to the strong westerly winds allowed the preservations of a very shallow katabatic flow (below 1.7 m above ground) close to the glacier surface, which is not captured by measurement sensors above. Furthermore, wind and turbulence

characteristics also infer a much stronger exposure of the central station to the across-glacier wind than the more sheltered margin station. Higher wind speeds at the central line appear to promote turbulent mixing close to the surface allowing the rush-in of high-speed fluid from the outer region into the near-surface atmospheric layer, as shown by Mott et al., (2016) for a wind tunnel experiment with warm air advection over a melting snow surface.

Turbulence measurements thus highlight the strong consequences of the development of across-glacier flows for the energy

balance at the glacier surface, although a thorough analysis of the origin of this flow requires a numerical modeling approach. The increasing wind velocity towards the centerline of the glacier promotes efficient heat exchange towards the glacier surface. Furthermore, measurements confirm that vertical heat fluxes measured below the jet height or in absence of the latter are significantly higher than measured at the jet height or just above where fluxes typically show its minimum. Turbulence in the layer above the wind speed maximum, as observed at the margin of the glacier, is largely decoupled from the flow below and

the underlying surface. Turbulence measured above the katabatic jet is thus no longer communicating with the surface (Denby 1999; Grachev et al., 2016; Mott et al., 2016). In case of the presence of an across-glacier flow, the very low-level wind speed maximum that potentially exist at the margin areas of the glacier might thus prevent heat exchange towards the glacier surface, partly decoupling the warmer air aloft. On the contrary, the higher low-level wind velocities at the more wind-exposed centerline and the associated increase in turbulence close to the surface might promote heat exchange towards the glacier

surface promoting ice melt there.

**5 Conclusion**

This study presents a unique set of turbulence data measured at a mid-latitude mountain glacier (Hintereisferner, Austria) during three weeks in summer 2018. The experiment was designed to capture near surface air flow dynamics and associated turbulent exchange processes at an along- and across-glacier transect. The high-density network of five meteorological stations

and eight turbulence sensors allowed us to investigate governing micrometeorological and turbulent heat exchange processes close to the glacier surface during both katabatic and non-katabatic dominated atmospheric flow conditions.

Measurements highlight the complex dynamics of boundary layer flows at a mountain glacier strongly affecting the local meteorology and glacier-atmosphere exchanges, with vertical profiles of wind speed and turbulent fluxes varying strongly for different flow conditions. We measured persistent low-level katabatic flows during daytime driving consistently cold air

temperatures close to the glacier surface with small spatial differences along the glacier. The across-glacier transect of stations showed katabatic jet height decreasing from a few meters at the centerline to less than two meters towards the glacier margin.



The spatial pattern of the katabatic flow across the glacier also induced a stronger spatial variability of near-surface air temperatures across the glacier than along the glacier. Turbulent heat exchange was especially driven by stronger wind velocities at the glacier centerline.

The measurement days analyzed showed a disturbance of the well-developed glacier wind by the evolution of an across-glacier flow induced by strong westerly winds. These predominantly westerly to northwesterly flows were associated with strong advection of heat with the larger scale flow. The horizontal heat advection was indicated by a significant rise in the near-surface air temperature which was greatest at the glacier margin. Local turbulence profiles of momentum and heat revealed a strong contribution of heat advection to the local heat budget. Strongest horizontal advection of heat was promoted by large

horizontal gradients of air temperature along the transect, coinciding with maximum heat exchange towards the glacier surface. The evolution of the across-glacier flow also coincided with an increasing turbulence from the peripheral zone towards the centerline. Turbulence measured along the across-glacier transect suggested different flow characteristics during disturbed situations between the peripheral zone and the centerline of the glacier. Profiles of momentum inferred a very low-level wind speed maximum below the lowest measurement level at the margin station potentially suppressing the heat exchange from the

higher atmospheric layers towards the glacier surface. In contrast, at the centerline of the glacier turbulence profiles suggested well-developed flow with high wind velocities promoting strong turbulence close to the glacier surface.

At the peripheral areas stronger exposure to the westerly winds might promote the preservation of a very shallow low-level katabatic jet which potentially decouples near-surface turbulence from higher atmospheric levels (Parmhed et al., 2004). At the centerline, westerly wind conditions coincided with an increase in low-level turbulent mixing and heat exchange towards

the glacier surface. In case of large-scale flows that are strong enough to disturb the katabatic wind on the glacier, we find the greatest increases in low-level heat exchange towards the glacier surface at the wind-exposed areas of the glacier, in our case at the centerline. This contrasts with previous studies (e.g. Sauter and Galos, 2016) that concluded heat exchange increases mostly at the peripheral areas of the glacier due to strongest heat advection. These earlier findings however, appear to be only valid for situations when the katabatic flow at the centerline of the glacier was preserved. Furthermore, the steepness of the

surrounding terrain plays a decisive role for the sheltering of peripheral areas from heat advection from the surrounding terrain. Steeper terrain might thus lead to a stronger sheltering of peripheral areas from a disturbance of the katabatic flow by larger-scale flows associated with strong winds and lateral heat advection.

Our experiments highlight the difficulty of experimentally characterizing the micro-meteorological conditions over glaciers and its potential effect on the energy balance of the glacier surface. Even flux profiles at multiple locations at the glacier

provide only local scale information and turbulence sensors only allow measurements at a certain distance away from the glacier surface. In the case of shallow katabatic jet formation, the vertical flux divergence is high and the knowledge of the exact local jet height is critically important for the interpretation of turbulence profiles. Turbulence measurements close to the jet height or even above will provide underestimated values of momentum and vertical heat fluxes not reflecting the turbulence characteristics at the glacier surface. These measurements do not necessarily provide meaningful information about heat

exchange through the atmospheric layer adjacent to the ice surface. Furthermore, the origin of the across-glacier flow and

differences of the exposure to strong westerly winds at different parts of the glacier could not be ascertained due to limited number of stations at higher elevations on the glacier and in the near-by surroundings. Numerical methods such as large eddy simulations are required to complement our experiments to investigate the dynamics of the across-glacier flow and its development. Although measurements suggested the impact of across glacier flows on the local energy balance to be non-575 negligible, the frequency of such flows at other glaciers is not known.

## 6 Data availability

Data used in this paper will be made available upon request to the first author.

## 7 Author Contributions

RM, IS and LN designed the field experiment and RM, IS, LN and JM conducted field experiments. RM and IS analysed the 580 data with contributions from JM. RM prepared the manuscript with contributions from all Co-authors,

## 8 Funding

The work was funded by Swiss National Science Foundation (Project: The sensitivity of very small glaciers to micrometeorology. P300P2_164644) and Austrian Science Fund (FWF) grant T781-N32.

## 9 Acknowledgments

We thank Alexander Kehl, Josh Chambers, Maximilian Kehl, Mark Smith, Tom Smith, Anna Wirbel and Zora Schirmeister for assisting during the field campaign.

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
