# Peer review of "Spatio-temporal flow variations driving heat exchange processes at a mountain glacier"

_The Cryosphere, 2020_

## Referee Comment (RC1) · Anonymous Referee #1 · 28 Apr 2020

This paper sets out to examine the interaction between katabatic and across-glacier glacier flows, and how this contributes to turbulent heat exchange. To my knowledge, this is one of the largest and best quality datasets exploring this complex interaction.

I found the writing compelling, but I'm not certain I would have reached all the same conclusions by looking at the data. Apart from computing fluxes, the quantitative analysis contained in this work did not extend much beyond correlation coefficients. Given the volume and complexity of the data and day-to-day variability, almost all figures presented could benefit greatly from more quantitative analyses. Because of this, it is not clear which conclusions are truly substantiated by their data and which are a product

of statistical outliers influencing qualitative analysis. This work could also benefit from more robust analysis of magnitudes and sources of possible uncertainties.

Specific comments:

-"Sensitivity analysis, however, shows that this increase is no considerable even when reducing the surface roughness by an order of magnitude." A number for what they deem "not considerable" would be helpful.

-The authors delineate wind regimes as "katabatic situations" and "disturbed situations". Although grammatically correct, I don't feel that "situation" is the best choice of words here. In the caption of Figure 3, the authors use "katabatic conditions" and "disturbed conditions", which feels more appropriate. As an alternate, I suggest "katabatic flows" and "disturbed flows".

-The authors state, "Following these observations, the position of the jet-speed maximum can be estimated by linear interpolation between two heights where momentum fluxes are measured (Grachev et al., 2016). This estimate assumes that the momentum flux decreases linearly, and can be applied confidently only if the jet maximum height happens to be between the two measurement levels." I understand that it won't work if the jet maximum height occurs outside of the two measurement levels, but this reads that they are confident that linear interpolation is appropriate (which they later state provides a crude estimate).

-"Flux footprints tend to be smaller during disturbed situations", although I don't see this from Figure 3. To my eyes, the areas enclosed (b) are larger than those enclosed in (a). My guess is that these are envelopes of the superposition of all footprints over the day, but I'm uncertain. Additionally, are these footprints of 80% flux contribution? More clarity here would be appreciated.

-"This extreme increase of wind speed with height is confirmed by preliminary numerical simulations (not shown)." It is unclear to me what these numerical simulations are
confirming. Two hypotheses are listed previously – is the numerical simulation confirming either of those? Or are the simulations simply confirming that this is possible (that the measurements are not faulty)? Wind shearing in excess of 15 m/s over only 55 cm is very significant for a mountain glacier. In either case, this is an opportunity to provide more detail and build a clearer physical picture of the dynamics at play.

-The authors should be more explicit with what they consider a strong correlation. "Sensible heat fluxes, however, show a strong correlation with the low-level wind speed during disturbed situations". Although not weak, I would argue that -0.42 and -0.47 aren't particularly strong correlations.

I'm not sure I follow the justification nor the implications for the analysis at the end of page 16.

-"During disturbed situations turbulence data showed small spatial difference of turbulent heat exchange at the across-glacier transect". The resulting scatters look similar, but is there any structure in plots of w'T' at TT3 vs w'T' at TT1?

-"Fluxes are particularly similar at TT1 and TT3 despite significantly higher air temperatures observed at TT1" How similar is "particularly similar"? Again, a scatter and more site-to-site analysis would aid this discussion.

-"In contrast to the margin station TT1 which shows similar correlations between air temperature and turbulent heat fluxes for both situations, the central station TT3 shows no correlation between air temperatures and heat fluxes". Although -0.2 and -0.21 are similar numbers, neither are strong correlations. One could also argue that 0.06 and 0.12 are similar numbers.

-"Figure 8 illustrates the advection of heat as a function of the deviation of the flow from the dominant katabatic flow direction" – this statement is backwards.

-Figure 8 and some of the following analyses are misleading. When wind direction isn't parallel with the station alignment, heat is no longer being advected between stations. Even if HA is calculated only using wind component V, U must be considered to determine the source of the heat advection. For example: if considering stations TT2 and TT1, if V = 1 m/s and U = 0 m/s, then it reasonable to assume heat is being advected from TT1 to TT2. If V= 1 m/s and U = 0.1 m/s, the source of the advection is slightly further up-glacier than TT1, so the measurement of HA is more inaccurate, as it assumes the up-glacier conditions are the same as those at TT1. This becomes a far more uncertain if V = 1 m/s and U = 5 m/s, for example. A clearer analysis of uncertainties and error here (and in figure 9) would be helpful. Currently, much of the information in Figure 8, along with the statement "Horizontal heat advection HA increased with temperature differences and V-component along the transect line" are guaranteed results considering that is how HA is defined. I wonder if factoring in these uncertainties would improve correlation coefficients between HA and w'T', as although 0.31 is a higher correlation than 0.19, I wouldn't call either of them a strong correlation.

The discussion could benefit from a rewrite with far more quantitative analysis to justify the interpretation of results, as it currently reads as purely qualitative. Some examples:

-"Second, the transect stations reveal a trend for both situations from more frequently measured positive and small momentum fluxes at the margin to larger and more frequently measured negative momentum fluxes at the central station." Distributions would be helpful in justifying this. I don't see this trend in the katabatic situation.

-"... higher flux divergence of turbulent heat fluxes during disturbed situations." If all of the scatter in y is projected onto a single line across the x-axis, do (a vs. d), (b vs. e), and (c vs. f) really look so different? How much higher are the flux divergences?

-"During westerly flow situations turbulence data at the centerline of the glacier (TT3) show a strong increase of downward vertical sensible heat fluxes with increasing downward momentum fluxes (negative values) (Fig. 10c)." This relationship is not apparent. I don't visually see any correlation between the colourbar (vertical sensible heat fluxes) with the y-axis (momentum fluxes).

-"While the mid-transect station TT2 evidences predominantly negative momentum fluxes with a considerably smaller flux divergence and smaller turbulent heat fluxes than observed at the centerline..." Certainly the maximum flux divergence is smaller, but how do the distributions/means compare? Is there any structure to the scatter plots? A similar analysis would be helpful in arguing that the turbulent heat fluxes are larger at the centerline. Is this comparison being done quantitatively or by eye?

Along a similar vein, some of the conclusions do not seem to fall from the work done in the paper.

-"Local turbulence profiles of momentum and heat revealed a strong contribution of heat advection to the local heat budget". Where was this done explicitly? The advective term is higher, but how strong is its contribution to the local heat budget (as a percentage, say)? What are the other components in the budget?

-"Strongest horizontal advection of heat was promoted by large horizontal gradients of air temperature along the transect, coinciding with maximum heat exchange towards the glacier surface." I'm not sure this is the conclusion that Figure 9 leads me to. At least in the case of TT2 & TT3, R(w'T',V)=0.56, but R(w'T',HA)=0.31. This implies to me that maximum heat exchange is more dependent on wind speed, but since HA = HA(V), elevated HA is somewhat correlated to elevated w'T', although is not the cause.  Again, performing an uncertainty analysis on HA given wind direction/speed could help make this distinction clearer.

-"Furthermore, the steepness of the surrounding terrain plays a decisive role for the sheltering of peripheral areas from heat advection from the surrounding terrain." Where does this conclusion come from?

Other aspects to tidy up:

-Occasionally, variables are not written in math mode/italicized (for example: w'T' on line 164, labels in all figures/tables).

- x'y' and \overline{x'y'} are used interchangeably, but should all be changed to the latter as they do not mean the same thing.

-Inconsistent labels on figures throughout (for example: "Height z (m)" & "Z (m)" are both used to denote height – Figure 5 even has both. Likewise with "wind speed U (m/s)" and "U (m/s)". Other labels such as (Fig 2 c,f) "Momentum, flux u'w' (K m sˆ{-1})" Contain all of these inconsistencies, an extra comma, and the wrong units).

-A comma instead of a period in "6,3 km" in line 86

-Throughout this paper, the figures are neither colourblind-friendly, nor are they B&W printer-friendly. They are also not saved in a .pdf format, so are low resolution. Figures 2, 4, and 5 are challenging to interpret as the colours appear very similar. Brown and grey, for example, are difficult to distinguish between. I would suggest a different colour palette and to make it consistent with Figure 3. -The dates of Figure 3 are not listed in chronological order.

-When appropriate, I would suggest making axis limits self-consistent. For example, Figure 4 (a&b), (c), (d), and (e) all have different x-axis limits. The same applies for Figure 4 (a/c) and (b/d) and Figure 9.

-I don't feel that diverging colourmaps are appropriate for the data presented in Figure 6, 9, or 10.

-Units need to be reviewed in all figures. To mention a couple: In Figure 6, $T_a/T_{mean}$ does not have units of C. Perhaps (C/C) is what is intended here. In Figure 8, (b) has incorrect units on the y-axis, and the x-axis has no units. Figure 10 has incorrect units on the y-axis and no units on the x-axis. Table 1 has units for RH but no other variables.

-The citations are not consistent with the journal's citation guide. Some journal names are cited in italic, and abbreviated journal names should have periods following them, i.e. "J Atmos Ocean Technol". The citations should be checked for consistency throughout. This journal is cited as both "Cryosphere" and other times "The

Cryosphere", not all journal titles are abbreviated appropriately, etc.

---

## Short Comment (SC1) · 17 May 2020

Thank you for this interesting study and this very nice dataset (although the time period is quite short). Have you examined the horizontal turbulent fluxes of heat (v'T' or u'T')? Since you observe significant variations of near-surface air temperature across the glacier, I guess this flux may be significant? Also: don't you think that adding a stability parameter (such as Ri or z/L) to the analysis would help to better understand the relationships between the turbulent heat flux and temperature or wind speed? Jean Emmanuel

---

## Referee Comment (RC2) · Jonathan Conway (Referee) · 27 May 2020

Review of manuscript "Insights into the effect of spatial and temporal flow variations on turbulent heat exchange at a mountain glacier" by Mott et al.

Jono Conway

**General comments**

This manuscript presents and discusses data from a novel field campaign conducted on the margin of the ablation area of a mountain glacier. Spatial patterns of near-surface turbulence along with wind speed and temperature are analysed during summer melt conditions for a series of case-study days with cross-glacier winds. The analyses aim to explore the interaction of advection, boundary layer structure and surface heat flux. The measurements are well designed, and the analyses presented draw a coherent picture of the interactions, despite the complexity of the many competing processing occurring. The limitations of the measurements (and indeed those in many previous studies) are recognised and discussed. While much of the analysis is exploratory and limited to scatter plots and linear correlations, plausible hypotheses are given for the patterns observed. These hypotheses should be addressed using large-eddy simulations in future studies to enable the mechanisms for the observed cross-glacier wind and its interaction with the down-glacier flow and turbulent heat fluxes to be analysed in greater depth. However, the quality and novelty of the data, the analysis presented and the hypotheses posed in the current paper is enough to warrant publication. While some of the conclusions are somewhat speculative, the paper makes an important contribution in that it highlights that the complexity of interactions between boundary layer and meso-scale dynamics in mountainous terrain limit the generalisation of results from specific locations to other glaciers, and that further efforts to measure and model boundary layers over mountain glaciers are needed if we are to properly understand the role of processes such as advection in models of glacier melt.

The manuscript would benefit from the addition of some context for the general meteorological conditions during campaign, especially timeseries of temperature and wind speed/ direction during the 5 selected days. This would provide the reader with a more intuitive introduction to the meteorology between relationships are discussed in later figures. These figures should also include an indication of time periods defined as 'katabatic' and 'disturbed' as this is unclear. In the discussion section, the authors should reflect further on the (potential) implications for measurements and modelling of turbulent heat fluxes, wind speed and air temperature distributions on other glaciers. Along with this the authors could provide more recommendations for future research.

Specific comments to improve the paper are provided below, but in general the paper is very well written, and figures well presented. My only concern with the analysis presented is the use of ratios to normalise temperature and wind speed in Figure 6, 7 and Table 1, and I would suggest the authors instead use anomalies (in K and ms$^{-1}$, respectively). This is especially important for temperature, where the fractional difference for the same change in temperature (in C) become smaller as daily mean temperature (in C) increases. If the authors wish to retain the current method, the theoretical basis for using ratios needs more explanation. The discussion of temperature differences between sites and situations is also very hard to compare with the current figures (see specific comments), but a change to anomalies and addition of timeseries of from each site should address this.

While the use of scatter plots makes it a little hard to interpret the density of data in certain figures, the ability to use colour warrants this approach. For some figures (Fig 9 and 10), histograms added along the x and y axes would enable the reader to see differences in the distribution that are discussed in the text (e.g. https://matplotlib.org/3.1.0/gallery/lines_bars_and_markers/scatter_hist.html).

In short, with some changes to clarify ambiguities of method and the presentation of additional results to support some statements, this manuscript will make a good addition to the literature.

**Specific comments:**

41 – the sensitivity of melt rate to air temperature is not only controlled by net longwave and turbulent heat flux, but also controlled by snowfall-albedo feedbacks – consider changing "controlled" to 'strongly affected' or similar.

48 – 'several studies' – worth adding additional references to this sentence or rewording.

49 – "near-surface warming" – it is unclear what is meant here – the katabatic models discussed in the previous sentence predict enhanced turbulent heat fluxes due to increased wind speed, not temperature. Please revise.

122 – please list the model numbers of the other instrumentation, including the young anemometers, the 2d sonic anemometer and the air temperature, rh and pressure sensors. Please also note if the t/rh sensors were passively or actively ventilated and if any corrections were made to raw data aside from the eddy-covariance data.

127 – it would be useful to expand further on the choice of 1-minute averaging period, as this departs significantly from often-used averaging periods of ~30 minutes. Perhaps present some of the analysis mentioned or comment on the effect of the short averaging period on, e.g. average heat fluxes.

147-155 – please clarify the criteria used to define katabatic vs disturbed conditions as there are several different versions given in this paragraph and the figure captions –

  o   i.e. did disturbed situation require wind shift from just W/NW or also E sector?
  o   please define whether 'time periods' on line 149 means 1-min or 30-min periods.
  o   Line 150 says that disturbed required WD shift of >50 degree over 30 mins, yet Figure 2 has many disturbed situations with average WD around 200 degrees?
  o   Figure 2 caption says katabatic required consistent WD during 30-min period – are there time periods that are excluded from the analysis as they do not fit either criteria?
  o   Are the data sub-set solely on one station (tt3), or classified individually based on WD at each station?
  o   Perhaps adding a timeseries of each case-study day, showing periods defined as katabatic and disturbed at TT3 would be useful.

223 – 'Flux footprints tend to be smaller during disturbed situations." Figure 3 shows a larger overall footprint area – perhaps worth clarifying that footprints for individual periods are smaller but the more varied orientation during disturbed conditions results in a larger overall footprint, if this is the case.

227 – Do you think the different instrumentation contributes significantly to the differences observed between level 3 and the lower two levels?

227 – Do you mean a secondary larger-scale wind system above level 2? If so, please clarify.

234 – "This extreme increase of wind speed with height is confirmed by preliminary numerical simulations (not shown)". As the reader cannot assess this without presenting the data, please remove or modify this sentence.

259 – 'higher streamwise momentum fluxes" please revise – I presume you mean "larger negative streamwise momentum fluxes"?

268 – 'on 2018-08-20' – I presume you mean on all case-study days? Please revise

277 – 'the temporal variability of flux profiles increased significantly for disturbed situations' – it is very hard to assess this statement from Figure 5 – please add further statistics to describe the mean and variability of the fluxes or reword.

Figure 6 – consider moving TT3 to the x axis on these plots as it is functioning here as a common variable (hence is more like the 'independent' variable).

Figure 6 – it is hard to assess the density of points in the scatter plot – consider using a transparency for the points so that more dense data shows as darker shades.

Figure 6 – the colour scale for disturbed conditions would be better to avoid white tones as the are hard to read. Scale used in Figure 9 would be better.

308-332 – there are many statements in this section at are not clearly supported by the data presented in Figure 6. The addition of timeseries of WD/WS and temperature from multiple sites would be of great benefit here.

310 – "significant increase in the near-surface air temperature of several degrees (Fig. 6d-f)" – this cannot be ascertained from the current figure 6 as the units are normalised. Please use anomalies as suggested in general comments section or provide additional results to support this statement.

314 – "Local air temperatures at the higher altitude station TT4 showed the lowest sensitivity to changes in wind direction at TT3." It is unclear how the data support this statement – please clarify and revise.

315 – "The katabatic flow seemed to persist at the higher altitude station TT4 when at the same time all transect stations already evidenced a westerly flow (Fig. 6b)." It is unclear how the data support this statement – please clarify and revise.

317 – "Air temperatures at the glacier tongue (WT1) appeared to be strongly affected by up-valley flows (Fig. 6f)." It is unclear how the data support this statement – please clarify and revise.

326 – "explain a larger spatial variability of the air temperature" – It is unclear how the data support this statement – please clarify and revise.

329 – Are the cooler temperatures during katabatic flows affected by diurnal changes in temperature? Ie. are katabatic conditions more common during cooler periods at night time?

Table 1 – what is $U_T$ ?

342 – 'all four turbulence stations' do you mean 'all three turbulence stations' or 'all 6 turbulence sensors'. Also please list what height data is from

361 – 'showed small spatial differences' – this is very hard to interpret from Figure 7 – a histogram of differences between fluxes at different stations would support this.

362 – "despite significantly higher air temperatures observed at TT1" – this is not shown and needs to be supported by additional results – perhaps a histogram of temperature differences between each site in different conditions.

388 – what fraction of periods were excluded?

Figure 8 – does this figure include all periods from the 5 days, or only disturbed periods? Please clarify in the caption. Please also add units and level used for HA calculation.

423 – "Similar to heat advection, peak vertical turbulent heat fluxes coincided with peak V-component at the centerline." - to what extent is this due to the correlation between mean wind speed and vertical fluxes? Please discuss.

424 – "Correlation coefficients $R(w'T',UT)$ were high between TT1-TT2 and TT2-TT3 station pairs with a slightly higher value for stations closer to the centerline." It is unclear how this relates to the data presented in Table 1. Please revise.

Figure 9 - consider adding histograms to each axis. It is currently very difficult to compare the distribution of points between different conditions and sites.

Figure 9 - consider adding histograms to the y axes. It is currently very difficult to compare the distribution of points between different conditions and sites.

509 - The steep moraine sides are likely to play a role in the sheltering of the site closest to the glacier margin, especially considering the sharp slope transitions and short distances involved. Thus, the flow hitting the glacier edge may not be well developed and still be affected by lee-side flow separation etc, reducing its ability to influence the stable glacier boundary layer. This may be worth discussing further here.

528 – as the study only presents data from 5 days, it would be more meaningful to say "during five days that displayed a distinct disruption of down-glacier flow during a three week period in summer 2018." Or similar.

541 – 'induced by strong westerly winds' – while this makes sense, the origin of the flow is still speculative so please revise.

552 – 'At the peripheral areas stronger exposure' – shouldn't this be 'weaker exposure'.

552 – As wind direction is not presented for TT1 it is impossible to assess if the 'preservation of a very-shallow low-level katabatic jet' is supported by the results. Figure 1 shows the WD is aligned at all levels at TT3 during disturbed situations – in order to support a katabatic jet at TT1 the wind direction would need to be maintained down-slope. The BL could still be decoupled at TT1 because of the strong thermal stratification, but this does not necessarily mean that a katabatic jet will exist at TT1. Please revise.

575 – "the frequency of such flows at other glaciers is not known" – this comment highlights that fact that the frequency of these flows has not been presented in the current study. This would be an easy and useful addition to the results.

**Editorial comments:**

16 – "the temporal change" -> "temporal changes"

121 – 'while as the fifth tower (WT1), with at these' –> 'while at the fifth tower (WT!), these'

125 – suggest changing 'methodology' to 'data processing'

141 – 'our dataset is not allowing us a' -> 'our dataset does not allow a'

145-157 – suggest moving this section to the end of section 2.2 so that it proceed mention of katabatic conditions at line 137.

160 – please add units for HA and FD

163 – what height wind speed was used for HA calculations?

188 – add 'for each case-study day' to caption.

225 - 'below 2.3 m above ground," -> 'below 2.3 m,'

275 – 'measurement levels turbulence' -> measurement levels, turbulence'

289 – '(TT1, TT2, TT3)' -> (TT1, TT3)

Table 1 – please add the sensor height or level used to the Table caption. Please add units for w'T'

371 – please add height or level used for heat flux

377 – please add units for HA

497 – 'supposed' – 'hypothesised'

560 – 'surrounding terrain' – 'surrounding terrain in this study'

---

## Author Comment (AC1) · 2 Jul 2020

**Referee Anonymous:**

**RC1 General comment:** This paper sets out to examine the interaction between katabatic and acrossglacier glacier flows, and how this contributes to turbulent heat exchange. To my knowledge, this is one of the largest and best quality datasets exploring this complex interaction. I found the writing compelling, but I'm not certain I would have reached all the same conclusions by looking at the data. Apart from computing fluxes, the quantitative analysis contained in this work did not extend much beyond correlation coefficients. Given the volume and complexity of the data and day-to-day variability, almost all figures presented could benefit greatly from more quantitative analyses. Because of this, it is not clear which conclusions are truly substantiated by their data and which are a product of statistical outliers influencing qualitative analysis. This work could also benefit from more robust analysis of magnitudes and sources of possible uncertainties.

General Response: we gratefully thank referee1 for his comments and suggestions to add a more quantitative analysis. We revised all figures additionally showing distributions of the data (see comments to referee 2). We also revised the text ensuring a clearer reference to numbers such as means, correlation coefficients or medians. We also avoid saying that something is a strong or weak correlation, and instead simply report the values in the text. We have added more quantitative analysis according to the specific suggestions or reviewer 2, but should you have particular additional analyses that you would like to see, then please specify those and we can include them where possible.

We revised figures 2, 3, 4, 5, 6, 7, 8, 9 and 10. All suggested figure improvements and adjustments have been made and checked for consistency. Used colours are now colour-blind friendly. Additionally, added three new figures:

- new Figure 2 describing the multi flux decomposition,
- new Figure 3 showing the time series of air temperature, wind and the classification scheme for katabatic and disturbed conditions.
- new figure 13 showing the divergences of the vertical and horizontal heat fluxes.

Furthermore, we added a new table showing estimates on flux footprint area and spitted the original table 1 in two tables – table 2 and table 3.

In the following we respond to all comments and provide the revised figures the responses are referring to at the end of the document.

**Specific comments:**

• "Sensitivity analysis, however, shows that this increase is no considerable even when reducing the surface roughness by an order of magnitude." A number for what they deem "not considerable" would be helpful.

**Response:** We have calculated the area enclosed by the footprints for the different conditions and stations. Increasing the roughness from 0.004 to 0.01 results in a decrease of footprint sizes that depends on the flow conditions, but is consistent between the stations. For katabatic flow the footprint size is 88 % of the original, and for the disturbed: 79 %. We now provide the areas of the footprint for two different roughness lengths:

Table 1: Estimates on flux footprint area in m2 for surface roughness of z0 = 0.004 m and z0 = 0.01 m. Flux footprint areas are provided for disturbed and katabatic flow conditions and for the three transect stations TT1, TT2 and TT3. With z0 = 0.004

|                   | TT1                  | TT2         | TT3                  |
|-------------------|----------------------|-------------|----------------------|
| katabatic         | $2.88 * 10^3$        | $2.31*10^3$ | $3.43*10^3$          |
| disturbed         | 6.35*10 3 | $6.5*10^3$  | 8.42*10 3 |
| With $z_0 = 0.01$ |                      |             |                      |
|                   | TT1                  | TT2         | TT3                  |

| katabatic | $2.5 * 10^3$ | $2.04*10^3$ | 3.03*10 3 |
|-----------|--------------|-------------|----------------------|
| disturbed | $5.01*10^3$  | $5.1*10^3$  | 6.67*10 3 |

(decrease of footprint size with increasing roughness between 0.004 and 0.01 is Katabatic: 88 % of the original, disturbed: 79 % )

- The authors delineate wind regimes as "katabatic situations" and "disturbed situations". Although grammatically correct, I don't feel that "situation" is the best choice of words here. In the caption of Figure 3, the authors use "katabatic conditions" and "disturbed conditions", which feels more appropriate. As an alternate, I suggest "katabatic flows" and "disturbed flows".
   Response: we now change the word situations to conditions throughout the manuscript.
  - The authors state, "Following these observations, the position of the jet-speed maximum can be
    estimated by linear interpolation between two heights where momentum fluxes are measured
    (Grachev et al., 2016). This estimate assumes that the momentum flux decreases linearly, and
    can be applied confidently only if the jet maximum height happens to be between the two
    measurement levels." I understand that it won't work if the jet maximum height occurs outside
    of the two measurement levels, but this reads that they are confident that linear interpolation is
    appropriate (which they later state provides a crude estimate).

**Response:** Indeed, the assumption that the momentum profile changes linearly with height is only rough and we use it here as the best guess to estimate of the jet height, following the study of Grachev et al. An independent study that is not part of this work, however, does show confidence in this estimate. Still, we have now changed the wording in the text to make it clear that this is generally indeed a rough estimate.

 "Flux footprints tend to be smaller during disturbed situations", although I don't see this from Figure 3. To my eyes, the areas enclosed (b) are larger than those enclosed in (a). My guess is that these are envelopes of the superposition of all footprints over the day, but I'm uncertain. Additionally, are these footprints of 80% flux contribution? More clarity here would be appreciated.

**Response:** Indeed, this was an imprecise formulation. The horizontal extent of the footprints for individual periods are smaller in disturbed conditions, however, the larger variability of wind direction during disturbed conditions results in an overall larger area of the climatological footprints. We now also provide more details on the footprint calculations and results.

• "This extreme increase of wind speed with height is confirmed by preliminary numerical simulations (not shown)." It is unclear to me what these numerical simulations are confirming. Two hypotheses are listed previously – is the numerical simulation confirming either of those? Or are the simulations simply confirming that this is possible (that the measurements are not faulty)? Wind shearing in excess of 15 m/s over only 55 cm is very significant for a mountain glacier. In either case, this is an opportunity to provide more detail and build a clearer physical picture of the dynamics at play.

**Response:** We thank the reviewer for spotting this inconsistency. There was a bug in processing the wind data that resulted in this unphysical result. The corrected analysis shows no strong increase in wind velocity between level 2 and 3. We apologize for this error. It, however, does not affect the remainder of the results as the error was only in the assimilation of the data from the 2D sonic. We have now also skipped the part of the text related to the preliminary numerical results as these are not yet ready for publication. These results are, however, in fact showing the presence of strong winds above the glacier.

• The authors should be more explicit with what they consider a strong correlation. "Sensible

heat fluxes, however, show a strong correlation with the low-level wind speed during disturbed situations". Although not weak, I would argue that -0.42 and -0.47 aren't particularly strong correlations. I'm not sure I follow the justification nor the implications for the analysis at the end of page 16.

**Response:** We agree that a correlation of 0.47 cannot be considered very strong. We have now recalculated correlation coefficients for wind velocity anomalies and also for new conditions defined for disturbed situations (we use a smaller wind sector that decreases the uncertainty due to flow not aligned with the transect). We revised the paragraph accordingly and no longer stating that something is a high correlation:

Turbulence data reveal higher vertical turbulent sensible heat fluxes during disturbed than during katabatic conditions. Higher heat fluxes coincide with higher air temperatures particularly at the margin station (Fig. 7 d-f). This is also reflected by a mean turbulent heat flux for disturbed conditions (-0.051 K m/s) being significantly higher than during katabatic conditions (-0.037). With the melting surface of the glacier at zero degrees, the increasing near-surface temperature gradients coincided with an increase of downward turbulent heat flux. As already mentioned, near-surface wind speeds during disturbed conditions were typically lower than the daytime average wind speed. Sensible heat fluxes, however, show a much higher correlation with the low-level wind speed (-0.5 and -0.62 for TT1 and TT3) during disturbed conditions than during katabatic flow conditions (-0.15 and -0.18 for TT1 and TT2). For disturbed conditions, no correlation between sensible heat flux and air temperature can be found (-0.001 and 0.16 for TT1 and TT3).

• "During disturbed situations turbulence data showed small spatial difference of turbulent heat exchange at the across-glacier transect". The resulting scatters look similar, but is there any structure in plots of w'T' at TT3 vs w'T' at TT1?

**Response**: We are now showing the structure of the data of vertical turbulent heat flux through histograms, which more clearly show the small differences between stations for disturbed conditions compared to katabatic conditions when TT3 shows higher fluxes than TT1 and TT2.

- "Fluxes are particularly similar at TT1 and TT3 despite significantly higher air temperatures observed at TT1" How similar is "particularly similar"? Again, a scatter and more site-to-site analysis would aid this discussion.
- "In contrast to the margin station TT1 which shows similar correlations between air temperature and turbulent heat fluxes for both situations, the central station TT3 shows no correlation between air temperatures and heat fluxes". Although -0.2 and -0.21 are similar numbers, neither are strong correlations. One could also argue that 0.06 and 0.12 are similar numbers.

**Response:** yes, we agree. We changed Figure 6 which now shows the distribution of temperature anomalies for all stations during katabatic and disturbed conditions. We further revised this section, which now reads: During disturbed conditions turbulence data showed small spatial differences of turbulent heat exchange at the across-glacier transect (Figure 7b). Fluxes are similar for all transect stations despite significantly higher air temperature anomalies observed at TT1 than at TT3 (+1.8°C for TT1 and +1.2°C for TT3; Figure 6b). While air temperatures were lower at TT3 than at TT1, higher wind velocities at the centreline appeared to promote heat exchange there (Figure 7b). This is also confirmed by statistics shown in Table 1. At the central station wind shows higher correlations with turbulent heat fluxes than at the margin station.

 "Figure 8 illustrates the advection of heat as a function of the deviation of the flow from the dominant katabatic flow direction" – this statement is backwards.
 Response: we revised this sentence.

• Figure 8 and some of the following analyses are misleading. When wind direction isn't parallel with the station alignment, heat is no longer being advected between stations. Even if HA is calculated only using wind component V, U must be considered to determine the source of the heat

advection. For example: if considering stations TT2 and TT1, if V = 1 m/s and U = 0 m/s, then it reasonable to assume heat is being advected from TT1 to TT2. If V= 1 m/s and U = 0.1 m/s, the source of the advection is slightly further up-glacier than TT1, so the measurement of HA is more inaccurate, as it assumes the up-glacier conditions are the same as those at TT1. This becomes a far more uncertain if V = 1 m/s and U = 5 m/s, for example. A clearer analysis of uncertainties and error here (and in figure 9) would be helpful. Currently, much of the information in Figure 8, along with the statement "Horizontal heat advection HA increased with temperature differences and V-component along the transect line" are guaranteed results considering that is how HA is defined. I wonder if factoring in these uncertainties would improve correlation coefficients between HA and w'T', as although 0.31 is a higher correlation than 0.19, I wouldn't call either of them a strong correlation.

**Response**: We agree that imperfect alignment with the transect would lead to partially erroneous conclusions of where the air is coming from. The conditions in which V = 1 m/s and U = 5 m/s would indeed mean that the along-transect component is negligible and the wind is coming from almost perpendicular direction to the transect. These kinds of conditions have now been a priory filtered out of our analysis as we only examine a sector that is more or less aligned with the transect when we examine heat advection. We therefore limited the analysis of heat advection and horizontal heat divergences between stations to a smaller wind sector of 60°. Beyond this, the uncertainty related to not perfectly aligned flows is hard to be quantified in a reliable way.

• "Second, the transect stations reveal a trend for both situations from more frequently measured positive and small momentum fluxes at the margin to larger and more frequently measured negative momentum fluxes at the central station." Distributions would be helpful in justifying this. I don't see this trend in the katabatic situation.

**Response**: we added distributions to Figure 10 which shows the shift of the curve towards more negative momentum fluxes and positive horizontal heat fluxes at TT3 than at TT1 (see below).

• higher flux divergence of turbulent heat fluxes during disturbed situations." If all of the scatter in y is projected onto a single line across the x-axis, do (a vs. d), (b vs. e), and (c vs. f) really look so different? How much higher are the flux divergences?

**Response**: we now directly compare the distributions of flux divergences during katabatic conditions against disturbed conditions in Figure 10 (shown later in this response document). This shows that we have a flatter distribution for FD during disturbed conditions with a higher value at the peak of the distribution.

 "During westerly flow situations turbulence data at the centerline of the glacier (TT3) show a strong increase of downward vertical sensible heat fluxes with increasing downward momentum fluxes (negative values) (Fig. 10c)." This relationship is not apparent. I don't visually see any correlation between the colourbar (vertical sensible heat fluxes) with the y-axis (momentum fluxes).

**Response**: we added now a new figure 12 showing this relationship between strong increase of downward vertical sensible heat fluxes with increasing downward momentum fluxes (negative values). We revised the text accordingly:

We are not only interested in changes in the turbulent structure when changing from katabatic to disturbed conditions but also on the effect of heat advection on the turbulent heat fluxes. Turbulence data of katabatic and disturbed conditions reveal some similarities along the transect stations but also pronounced differences between the different flow conditions (Fig. 10 a, b). First, the three transect stations show a similar trend for both conditions with an increase of the vertical turbulent heat flux (Fig. 7 a, b) and heat flux divergence (Figure 10 a, b) from the margin station towards the central station. Second, the transect stations reveal a trend for both conditions from more frequently measured positive and small momentum fluxes at the margin to larger and more frequently measured negative momentum fluxes at the central station (Fig. 10 a, b). On the other side, the largest differences are the much higher magnitudes of turbulent fluxes of momentum and heat as well as higher flux divergence of turbulent heat fluxes during disturbed conditions.

In order to assess the effect of heat advection on the heat exchange processes during disturbed conditions we focus our analysis on flow characteristics during those conditions (Fig. 10; Fig. 12). During westerly flow conditions turbulence data at the centreline of the glacier (TT3) show a strong increase of downward vertical sensible heat fluxes with increasing downward momentum fluxes (negative values) (Fig. 12b). The strongest vertical turbulent heat fluxes coincided with peak vertical heat divergence (Fig. 10 d). At the more wind-exposed centreline, negative momentum fluxes and the strong vertical heat flux divergence (Fig. 10 b) indicate that no pronounced katabatic jet is present below the lowest measurement level and that measurements were conducted within a stable atmospheric layer with increasing wind velocities with height featuring strong flux gradients close to the surface. Strong turbulent momentum and sensible heat fluxes combined with strong flux divergence at TT3 suggest very efficient turbulence transfer towards the surface in case of advection.

"While the mid-transect station TT2 evidences predominantly negative momentum fluxes with a considerably smaller flux divergence and smaller turbulent heat fluxes than observed at the centerline: : :" Certainly the maximum flux divergence is smaller, but how do the distributions/means compare? Is there any structure to the scatter plots? A similar analysis would be helpful in arguing that the turbulent heat fluxes are larger at the centerline. Is this comparison being done quantitatively or by eye? Along a similar vein, some of the conclusions do not seem to fall from the work done in the paper.

**Response:** we revised Figure 10 now showing the structure of the data by presenting the histograms. This figure supports the statement that the mid-transect station TT2 evidences predominantly negative momentum fluxes with a considerably smaller flux divergence and smaller turbulent heat fluxes than observed at the centreline. It also shows the strong differences in the flux divergence between katabatic and disturbed conditions.

• "Local turbulence profiles of momentum and heat revealed a strong contribution of heat advection to the local heat budget". Where was this done explicitly? The advective term is higher, but how strong is its contribution to the local heat budget (as a percentage, say)? What are the other components in the budget?

**Responds:** we now also show the horizontal flux divergence calculated only for the narrow wind sector of 250°-290° which ensures that the flow was aligned with the transect. This figure shows that both horizontal and vertical flux divergences are at the same order of magnitude but the vertical heat flux divergence is larger, in particular at the central station.

"Strongest horizontal advection of heat was promoted by large horizontal gradients of air temperature along the transect, coinciding with maximum heat exchange towards the glacier surface." I'm not sure this is the conclusion that Figure 9 leads me to. At least in the case of TT2 & TT3, R(w'T',V)=0.56, but R(w'T',HA)=0.31. This implies to me that maximum heat exchange is more dependent on wind speed, but since HA = HA(V), elevated HA is somewhat correlated to elevated w'T', although is not the cause. Again, performing an uncertainty analysis on HA given wind direction/speed could help make this distinction clearer.

**Responds:** We have now limited our analysis of heat advection to a narrow 60° wind sector.**

• "Furthermore, the steepness of the surrounding terrain plays a decisive role for the sheltering of peripheral areas from heat advection from the surrounding terrain." Where does this conclusion come from?

**Response:** This conclusion is based on the analysis of turbulence data profiles suggesting less developed flow at the peripheral areas during disturbed conditions compared to the central stations. The sign of momentum and horizontal heat fluxes suggest the presence of a very low-level jet below the lower measurement height at TT1, but a well-developed flow at TT3. We updated the discussion to have a more in-depth discussion of the sheltering effect: *The topographic setting which is typical for alpine glaciers are likely to play a significant role in the sheltering of the site closest to the glacier margin. Steep moraine sides and sharp slope transitions at the glacier margin strongly affect the local boundary layer flow (i.e. lee-side flow separation) reducing the ability of the flow hitting the glacier edge to influence the stable glacier boundary layer. Contrary, well developed flows at the glacier line and associated higher wind speeds appear to promote turbulent mixing close to the surface allowing the rush-in of high-speed fluid from the outer region into the near-surface atmospheric layer, as shown by Mott et al., (2016) for a wind tunnel experiment with warm air advection over a melting snow surface.*

**Other aspects to tidy up:**

• Occasionally, variables are not written in math mode/italicized (for example: w'T' on line 164, labels in all figures/tables).

**Response:** Yes, we agree and revised all figures accordingly showing the correct in math mode labels.

• x'y' and noverline{x'y'} are used interchangeably, but should all be changed to the latter as they do not mean the same thing.

**Response**: we agree with the referee. We revised all figures showing the correct in math mode labels including overbars.

Inconsistent labels on figures throughout (for example: "Height z (m)" & "Z (m)" are both used to denote height – Figure 5 even has both. Likewise with "wind speed U (m/s)" and "U (m/s)". Other labels such as (Fig 2 c,f) "Momentum, flux u'w' (K m s1-1)" Contain all of these inconsistencies, an extra comma, and the wrong units).

**Response:** we revised all figures and ensured consistency.

• A comma instead of a period in "6,3 km" in line 86 **Response:** we revised this.

• Throughout this paper, the figures are neither colourblind-friendly, nor are they B&W printer-friendly. They are also not saved in a .pdf format, so are low resolution. Figures 2, 4, and 5 are challenging to interpret as the colours appear very similar. Brown and grey, for example, are difficult to distinguish between. I would suggest a different colour palette and to make it consistent with Figure 3. -The dates of Figure 3 are not listed in chronological order.

**Response:** we revised all figures including color schemes and legends.

• When appropriate, I would suggest making axis limits self-consistent. For example, Figure 4 (a&b), (c), (d), and (e) all have different x-axis limits. The same applies for Figure 4 (a/c) and (b/d) and Figure 9.

**Response**: we revised all figures ensuring self-consistent x- and y axis.

- I don't feel that diverging colourmaps are appropriate for the data presented in Figure
- 6, 9, or 10.

**Response:** we revised all figures including color schemes and legends.

• Units need to be reviewed in all figures. To mention a couple: In Figure 6, T\_a/T\_mean does not have units of C. Perhaps (C/C) is what is intended here. In Figure 8, (b) has incorrect units on the y-axis, and the x-axis has no units. Figure 10 has incorrect units on the y-axis and no units on the x-axis. Table 1 has units for RH but no other variables.

**Response**: we revised all figures including color schemes, legends, labels and units. We also changed  $T_a/T_mean$  to anomalies allowing a better physical interpretation including units.

• The citations are not consistent with the journal's citation guide. Some journal

names are cited in italic, and abbreviated journal names should have periods following them, i.e. "J Atmos Ocean Technol". The citations should be checked for consistency throughout. This journal is cited as both "Cryosphere" and other times "The Cryosphere", not all journal titles are abbreviated appropriately, etc.

**Response:** we revised the citations style to be consistent with the journal's citations style.

Figure: Multi-resolution flux decomposition of buoyancy flux as a function of time scale t for the four examined stations.

---

## Author Comment (AC2) · 2 Jul 2020

Respond to Jean Emmanuel

Thank you for this interesting study and this very nice dataset (although the time period is quite short). Have you examined the horizontal turbulent fluxes of heat (v'T' or u'T')? Since you observe significant variations of near-surface air temperature across the glacier, I guess this flux may be significant?

Also: don't you think that adding a stability parameter (such as Ri or z/L) to the analysis would help to better understand the relationships between the turbulent heat flux and temperature or wind speed?

**Response:** Dear Jean Emmanuel, thanks for your comments. Yes, the time period is quite short but those measurements were associated with a lot of effort which we could not afford for longer time period.

Yes, horizontal turbulent fluxes are significant and are larger than the vertical fluxes. When looking at the fluxes we always use the rotated flux which is called streamwise flux. That is why for disturbed conditions u'T' which is along the transect shows the same tendencies for the stations as does u'w' suggesting that we are most probably above a local jet height at TT1 and below it at TT2 and TT3.

[Figure]

Figure 11: Streamwise horizontal turbulent heat flux plotted against streamwise momentum flux for stations TT1, TT2 and TT3 (a). Vertical turbulent heat flux plotted against streamwise momentum flux for stations TT1, TT2 and TT3 (b). Data are only shown for disturbed conditions and the 60°wind sector from 240° to 300°.

We also add here a plot showing vertical heat flux against horizontal heat flux u'T' for disturbed and katabatic conditions. The horizontal heat flux is larger for katabatic flows than for disturbed ones.

[Figure]

We now also show the horizontal flux divergence calculated only for the narrow wind sector of 250°-290° which ensures that the flow was aligned with the transect. This figure shows that both horizontal and vertical flux divergences are at the same order of magnitude but the vertical heat flux divergence is larger, in particular at the central station.

[Figure]

Figure 13: Kernel distribution of horizontal (hDF) and vertical (vDF) heat flux divergence shown only for disturbed situations and the wind sector 240° to 300°.

We analysed stability parameter z/L and plot it against the vertical heat flux, We can detect a tendency of higher turbulent heat fluxes for weaker stability (i.e. during disturbed flows that are more near-neutrally stratified). The decrease of stability during disturbed flows is associated with higher wind speeds and therefore higher friction velocity.

We added a figure describing the stability parameter to the paper:

[Figure]

*Figure 7: Vertical heat flux plotted against anomalies of wind speed from mean daytime wind speed shown for stations TT1 –TT3 for katabatic situations a) and disturbed situations (b). Vertical momentum flux plotted against anomalies of wind speed from mean daytime wind speed (c) and against anomalies of air temperature from mean daytime air temperature (d) shown for station TT3 for katabatic situations and disturbed situations. Logarithm of Stability parameter z/L plotted against Vertical heat flux (e) and normalized wind speed (f) measured at TT3 during katabatic and disturbed flows.*

---

## Author Comment (AC3) · 2 Jul 2020

**Response to Referee 2 - Jono Conway:**

General Response: We thank Jono Conway for his very valuable comments which helped significantly to improve the representation of the results and the informative value of the figures. We revised figures 2, 3, 4, 5, 6, 7, 8, 9 and 10. All suggested figure improvements and adjustments have been made and checked for consistency. Used colours are now colour-blind friendly. Additionally, added three new figures:

- new Figure 2 describing the multi flux decomposition,
- new Figure 3 showing the time series of air temperature, wind and the classification scheme for katabatic and disturbed conditions.
- new figure 13 showing the divergences of the vertical and horizontal heat fluxes.

Furthermore, we added a new table showing estimates on flux footprint area and spitted the original table 1 in two tables – table 2 and table 3.

In the following we respond to all comments and provide the revised figures the responses are referring to at the end of the document.

**General comments**

 The manuscript would benefit from the addition of some context for the general meteorological conditions during campaign, especially timeseries of temperature and wind speed/ direction during the 5 selected days. This would provide the reader with a more intuitive introduction to the meteorology between relationships are discussed in later figures. These figures should also include an indication of time periods defined as 'katabatic' and 'disturbed' as this is unclear.

**Response**: We agree and have now added the time series of temperature anomalies, the wind velocity, deviation of wind direction from 200° (prevailing Katabatic wind direction) and the classification between katabatic and disturbed flows for two of the measurement days to demonstrate conditions during katabatic and disturbed situations. We decided not show the data for all five days as this would make the figure too unclear.

2. In the discussion section, the authors should reflect further on the (potential) implications for measurements and modelling of turbulent heat fluxes, wind speed and air temperature distributions on other glaciers. Along with this the authors could provide more recommendations for future research.

**Response**: Yes, we agree with the referee and will add a discussion on implications for modelling an measuring the distribution of turbulent heat fluxes, temperature and wind speed. A postdoc based at Innsbruck is currently doing 240m and 48m resolution LES simulations of these days with WRF which will allow us to include some specific experiences relevant to combining and comparing measurements such as these with modelling efforts. In terms of future research, we now include lessons learned from our instrumental campaign and some specific research goals we would want to explore in a follow up campaign if funding were available.

3. Specific comments to improve the paper are provided below, but in general the paper is very well written, and figures well presented. My only concern with the analysis presented is the use of ratios to normalise temperature and wind speed in Figure 6, 7 and Table 1, and I would suggest the authors instead use anomalies (in K and ms-1, respectively). This is especially important for temperature, where the fractional difference for the same change in temperature (in C) become smaller as daily mean temperature (in C) increases. If the authors wish to retain the current method, the theoretical basis for using ratios needs more explanation. The discussion of

temperature differences between sites and situations is also very hard to compare with the current figures (see specific comments), but a change to anomalies and addition of timeseries of from each site should address this. While the use of scatter plots makes it a little hard to interpret the density of data in certain figures, the ability to use colour warrants this approach. For some figures (Fig 9 and 10), histograms added along the x and y axes would enable the reader to see differences in the distribution that discussed are in the text (e.g. https://matplotlib.org/3.1.0/gallery/lines bars and markers/scatter hist.html).

In short, with some changes to clarify ambiguities of method and the presentation of additional results to support some statements, this manuscript will make a good addition to the literature.

**Response:** We followed Jono Conway suggestion to use anomalies and adapted Figure 6 and 7, as well as statistics shown in table 1. We further followed the suggestion to add histograms to figures 7, 9 and 10 which much better describe the distribution of the data and allows a better comparison between the data (i.e. between station or flow types). Figures are shown further below under specific comments.

**Specific comments:**

• 41 – the sensitivity of melt rate to air temperature is not only controlled by net longwave and turbulent heat flux, but also controlled by snowfall-albedo feedbacks – consider changing "controlled" to 'strongly affected' or similar.

**Response:** yes, we agree. We replace "controlled" by "strongly affected".

• 48 – 'several studies' – worth adding additional references to this sentence or rewording. **Response:** we reworded the sentence.

• 49 – "near-surface warming" – it is unclear what is meant here – the katabatic models discussed in the previous sentence predict enhanced turbulent heat fluxes due to increased wind speed, not temperature. Please revise.

**Response:** to be clear that these are different processes we changed the corresponding sentences to read: *Zhong and Whiteman (2008) claim that near-surface warming induced by katabatic flow could also be caused by along-slope warm-air advection, while Pinto et al., (2006) identify the entrainment of potentially warmer air down to the surface driven by stronger turbulent mixing. Furthermore, some studies highlighted the effect of katabatic flows in laterally decoupling the local atmosphere from its surrounding, thus lowering the climatic sensitivity of glaciers to external temperature changes (Shea and Moore, 2010; Sauter and Galos, 2016; Mott et al., 2019).*

122 – please list the model numbers of the other instrumentation, including the young anemometers, the 2d sonic anemometer and the air temperature, rh and pressure sensors. Please also note if the t/rh sensors were passively or actively ventilated and if any corrections were made to raw data aside from the eddy-covariance data.

**Response:** yes, we add the model numbers and the information that the RH/T sensor was actively ventilated.

127 – it would be useful to expand further on the choice of 1-minute averaging period, as this
departs significantly from often-used averaging periods of ~30 minutes. Perhaps present some of
the analysis mentioned or comment on the effect of the short averaging period on, e.g. average
heat fluxes.

**Response**: We have now added a plot showing the results of the MRD which highlights our choice of the averaging time, and have expended the text to provide more information.

The turbulence data were processed as follows: multi-resolution flux decomposition (MRD) was used to determine the optimal averaging time for the turbulence data (Vickers and Mahrt, 2003). MRD works as a wavelet transform that decomposes the signal into dyadic scales while preserving Reynolds averaging rules. The appropriate averaging time is usually taken to be that time scale at which the contribution to the flux (at its inter-quantile ranges) first crosses over zero (Vickers and Mahrt, 2003). The MRD analysis of the heat flux for the four examined stations during the period of the campaign (Figure 2) shows that due to its stable nature, the dominant turbulent contribution to the flux comes at scales smaller than 1 min, while the scales larger than 1 min already show a strong contribution to the turbulent flux up until a 5 min scale. Following the approach of Vickers and Mahrt (2003) however, we choose the appropriate averaging time scale to be that where the upper quantile crosses over zero, for comparability reasons we therefore block average the data from all stations with an averaging time of 1 minute

• 147-155 – please clarify the criteria used to define katabatic vs disturbed conditions as there are several different versions given in this paragraph and the figure captions – i.e. did disturbed situation require wind shift from just W/NW or also E sector?

**Response:** disturbed situation also include flow from easterly sector, but these were very rare. Now the analysis of horizontal heat flux and horizontal heat flux divergence was limited to the small wind sector of 290°+/- 30° which is flow along the transect (see revised methods below). This is also indicated in Figure 2 and 8. We added the following criteria: The classification varies for averaging time periods of 1 and 30 minutes. The data analysis based entirely on 1-minute averages used the following classification (as applied in Figure 2): (1) Pure katabatic conditions are defined as flows with persistent flow direction from southwest (defined as 200° at station TT3) and wind velocities larger than 3 m/s. (2) Disturbed conditions are defined by a deviation of wind direction of more than  $40^{\circ}$  and less than  $100^{\circ}$  from the dominant katabatic flow direction. This limits the flow sector to +/-  $30^{\circ}$ from the flow perfectly aligned with transect. Following these criteria, the analysis of turbulence data was performed for the following five days: 4, 5, 11, 15 and 20 August (referred to as day 1-5). During these days, persistent katabatic flow was disturbed by westerly winds or up-valley flows (strong shift of the dominant wind direction during the day from southwest to the westerly or easterly wind sector). Data used for the 30 minute-averaged profiles were classified using the following criteria: Pure katabatic flows are defined as flows with persistent flow direction from southwest (defined as 200° at station TT3) and wind velocities larger than 3 m/s for the entire 30-minute averaging time period. All other flows were classified as disturbed flows without lower limit of wind direction. Note that the upper turbulence sensor (CSAT, level 2) at TT2 was not working until 7 August, due to a faulty cable which had to be replaced. During this period turbulence profiles were analyzed for stations TT1 and TT3.

• please define whether 'time periods' on line 149 means 1-min or 30-min periods. **Response:** it refers to each 1-minute average.

- Line 150 says that disturbed required WD shift of >50 degree over 30 mins, yet Figure 2 has many disturbed situations with average WD around 200 degrees?
- Figure 2 caption says katabatic required consistent WD during 30-min period are there time periods that are excluded from the analysis as they do not fit either criteria?

• Are the data sub-set solely on one station (tt3), or classified individually based on WD at each station?

**Response**: the subsets are classified based on the wind direction and velocity measured at TT3. This has now been more clearly stated in the text and the deviations for stations other than TT3 were discussed (see above).

• Perhaps adding a timeseries of each case-study day, showing periods defined as katabatic and disturbed at TT3 would be useful.

**Response**: we fully agree that as it is stated in the text is confusing. We revised the method section to be clearer about the two different classification schemes used for analysis based on 1-minute averages and on 30-minute averages. Two slightly different schemes are used to allow a stricter classification for smaller averaging times as we expect more homogenous flow conditions during shorter time periods. We further show a time series now demonstrating the stricter classification scheme (see above).

• 223 – 'Flux footprints tend to be smaller during disturbed situations." Figure 3 shows a larger overall footprint area – perhaps worth clarifying that footprints for individual periods are smaller but the more varied orientation during disturbed conditions results in a larger overall footprint, if this is the case.

**Response:** Yes, we agree and added this to the text and have also added the information on the actual area enclosed by the footprints for the different conditions. We have also calculated the area enclosed by the footprints for the different conditions. Increasing the roughness from 0.004 to 0.01 results in a decrease of footprint sizes that depends on the flow conditions, but is consistent between the stations. For katabatic flow the footprint size is 88 % of the original, and for the disturbed: 79 %. We now provide the areas of the footprint for two different roughness lengths:

Table 1: Estimates on flux footprint area in m2 for surface roughness of z0 = 0.004 m and z0 = 0.01 m. Flux footprint areas are provided for disturbed and katabatic flow conditions and for the three transect stations TT1, TT2 and TT3. With z0 = 0.004

|                   | TT1                  | TT2                 | TT3                  |
|-------------------|----------------------|---------------------|----------------------|
| katabatic         | $2.88 * 10^3$        | $2.31*10^3$         | $3.43*10^3$          |
| disturbed         | 6.35*10 3 | $6.5*10^3$          | 8.42*10 3 |
| With $z_0 = 0.01$ |                      |                     |                      |
|                   | TT1                  | TT2                 | TT3                  |
| katabatic         | $2.5 * 10^3$         | $2.04*10^3$         | $3.03*10^3$          |
| disturbed         | $5.01*10^{3}$        | 5 1*10 3 | $6.67*10^3$          |

(decrease of footprint size with increasing roughness between 0.004 and 0.01 is Katabatic: 88 % of the original, disturbed: 79 %)

• 227 – Do you think the different instrumentation contributes significantly to the differences observed between level 3 and the lower two levels?

**Response**: We have now skipped this part of the text as wind data in the original version had some errors. Now, there is no strong increase in wind velocity between level 2 and 3.

**• 227 – Do you mean a secondary larger-scale wind system above level 2? If so, please clarify.**

**Response**: This part is skipped from the text as wind data in the original version had some errors. Now, there is no strong increase in wind velocity between level 2 and 3.

• 234 – "This extreme increase of wind speed with height is confirmed by preliminary numerical simulations (not shown)". As the reader cannot assess this without presenting the data, please remove or modify this sentence.

**Response:** we agree, we removed this sentence.

• 259 – 'higher streamwise momentum fluxes" please revise – I presume you mean "larger negative streamwise momentum fluxes"?

**Response:** yes, we agree, we revised this part accordingly.

• 268 – 'on 2018-08-20' – I presume you mean on all case-study days? Please revise **Response**: thanks! Yes, this is true – we revised it.

• 277 – 'the temporal variability of flux profiles increased significantly for disturbed situations' – it is very hard to assess this statement from Figure 5 – please add further statistics to describe the mean and variability of the fluxes or reword.

**Response**: we have to admit that this is not very clear and removed this sentence.

• Figure 6 – consider moving TT3 to the x axis on these plots as it is functioning here as a common variable (hence is more like the 'independent' variable).

**Response**: we revised figure 6 now showing kernel distributions for all stations for katabatic and disturbed conditions. We revised the text accordingly.

- Figure 6 it is hard to assess the density of points in the scatter plot consider using a transparency for the points so that more dense data shows as darker shades.
   Response: Please see comment above.
  - Figure 6 the colour scale for disturbed conditions would be better to avoid white tones as the are hard to read. Scale used in Figure 9 would be better.

**Response:** please see comment above.

• 308-332 – there are many statements in this section at are not clearly supported by the data presented in Figure 6. The addition of timeseries of WD/WS and temperature from multiple sites would be of great benefit here.

**Response:** we are now showing the time series of 2 days, for stations TT1 and TT3. We now also show the mean temperature anomaly for each station and condition.

 310 – "significant increase in the near-surface air temperature of several degrees (Fig. 6d-f)" – this cannot be ascertained from the current figure 6 as the units are normalised. Please use anomalies as suggested in general comments section or provide additional results to support this statement.

**Response**: we now show the anomalies indicating the change in temperature which provides a clearer picture.

- 314 "Local air temperatures at the higher altitude station TT4 showed the lowest sensitivity to changes in wind direction at TT3
- ." It is unclear how the data support this statement please clarify and revise.

**Response**: this is shown by the smallest temperature anomaly for disturbed flow. We revised the text accordingly:

Local air temperatures at the higher altitude station TT4 showed the lowest sensitivity to changes in wind direction at TT3, which is reflected by the smallest mean temperature anomaly for disturbed flows. Wind direction data at TT4 (not shown) suggest that the katabatic flow seemed to persist at the higher-altitude station TT4 when at the same time all transect stations already evidenced a westerly flow. Data thus suggest that the station TT4 was more sheltered from westerly flows than stations located at lower parts of the glacier.

• 315 – "The katabatic flow seemed to persist at the higher altitude station TT4 when at the same time all transect stations already evidenced a westerly flow (Fig. 6b)." It is unclear how the data support this statement – please clarify and revise.

**Response:** Yes, it is true the figure 6 does not show this because we always show the wind direction deviation based on TT3 measurements. As we want to stick with that we changed the sentence now to: Wind direction data at TT4 (not shown) suggest that the katabatic flow seemed to persist at the higheraltitude station TT4 when at the same time all transect stations already evidenced a westerly flow.

• 317 – "Air temperatures at the glacier tongue (WT1) appeared to be strongly affected by upvalley flows (Fig. 6f)." It is unclear how the data support this statement – please clarify and revise.

**Response:** we removed this sentence.

• 326 – "explain a larger spatial variability of the air temperature" – It is unclear how the data support this statement – please clarify and revise.

**Response**: We agree with the Referee that spatial differences are quite similar between the two flow conditions. We therefore decided to remove this sentence.

 329 – Are the cooler temperatures during katabatic flows affected by diurnal changes in temperature? Ie. are katabatic conditions more common during cooler periods at night time?
 Response: Our analysis is only focused on the daytime hours, as mentioned in the text, and therefore we

only examine daytime temperatures. We do also observe katabatic flows in the afternoon – we can therefore not link cooler air temperatures to diurnal changes.

• Table 1 – what is UT ?

**Response**: We thank the referee for detecting this inconsistency – UT is named V elsewhere in the text - wind velocity component along the transect V (wind speed component along the Transect)

 342 – 'all four turbulence stations' do you mean 'all three turbulence stations' or 'all 6 turbulence sensors'. Also please list what height data is from

**Response**: We revised the text now referring to three across glacier transect stations. We also added a more detailed list of heights etc.

Each tower measured wind properties at three heights above the ice surface (1.7 m (level 1) and 2.35 m (level 2) and 2.9 m (level 3)), as well as air temperature, relative humidity and pressure at level 1. The temperature and humidity sensors were actively ventilated. At the four turbulence towers (TT1-TT4) the wind sensors at level 1 and 2 were Campbell CSAT3 sonic anemometers, sampling at a frequency of 20 Hz, while as the fifth tower (WT1), with at these levels was recorded with two Young anemometers. At all towers the level 3 wind sensor was a two-dimensional sonic anemometer. Air temperature, relative humidity and air pressure was measured at each station at measurement level 1 with a 1-minute resolution.

 showed small spatial differences' – this is very hard to interpret from Figure 7 – a histogram of differences between fluxes at different stations would support this.

**Response**: we now add histograms to Figures 7. The distributions nicely show that spatial differences of turbulent heat fluxes are particularly small for disturbed flows and are higher for katabatic flows.

• 362 – "despite significantly higher air temperatures observed at TT1" – this is not shown and needs to be supported by additional results – perhaps a histogram of temperature differences between each site in different conditions.

**Response**: a histogram of air temperatures is now shown in Figure 7. Additionally, mean anomalies of air temperature are given for TT1 and TT3 showing higher air temperature anomalies at TT1 for disturbed situations.

• Figure 8 – does this figure include all periods from the 5 days, or only disturbed periods? Please clarify in the caption. Please also add units and level used for HA calculation.

**Response:** we revised Figure8 accordingly.

• 423 – "Similar to heat advection, peak vertical turbulent heat fluxes coincided with peak Vcomponent at the centerline." - to what extent is this due to the correlation between mean wind speed and vertical fluxes? Please discuss.

**Response:** we revised the discussion of correlations between HA, wind and vertical turbulent heat fluxes: We are interested in the efficiency of the horizontal heat transport to warm near-surface air layers and thus to indirectly promote turbulent heat exchange towards the ice surface contributing to the surface energy balance. We therefore analyzed the relationship between horizontal heat advection HA (TT1-TT2 and TT2-TT3), the vertical turbulent heat flux and the V-component along the transect, illustrated in Fig. 9. Additionally, correlation coefficient R between those variables are provided (Table 1). Note that for this analysis we considered only data for the 60° wind sector (see methods, disturbed conditions). Consistent with small correlations between air temperature and  $\overline{w'T'}$ , correlations between HA and  $\overline{w'T'}$  are rather small for all stations. Highest correlation was found at TT3 (0.31). Peak vertical turbulent heat fluxes coincided with peak V-component at the centreline. Correlation coefficients  $R_{(wT,UT)}$  were higher between TT2 and TT3 (0.56). Turbulent heat fluxes showed slightly smaller mean values at TT1 (Figure 9b), coinciding with significantly smaller wind speeds (Figure 9a). Furthermore, the correlation between wind speed and vertical turbulent heat flux at the peripheral station was smaller (-0.5) than at the centreline (-0.62). Thus, at the centerline (TT3) strong winds not only promote stronger heat advection (Figure 9a) but also promote maximum downward turbulent heat exchange (Figure 9b). Heat advection appears to enhance turbulent heat exchange towards the glacier surface by enhancing near-surface temperature gradients. Consequently, at the glacier centreline (TT3) stronger winds enhance both the heat advection and the turbulent heat exchange.

• 424 – "Correlation coefficients R(w'T',UT) were high between TT1-TT2 and TT2-TT3 station pairs with a slightly higher value for stations closer to the centerline." It is unclear how this relates to the data presented in Table 1. Please revise.

**Response:** please see the revised text above.

• Figure 9 - consider adding histograms to each axis. It is currently very difficult to compare the distribution of points between different conditions and sites.

**Response**: we added histograms to Figures 9 and 10. We now show heat advection as a function of V component and vertical heat flux. Showing both stations in one plot allows a much better comparison of

the distribution of the data. In Figure 10 we now present all stations in one plot. Panel c additionally presents the data from station TT3 but for katabatic and disturbed situations to allow a direct comparison.

509 - The steep moraine sides are likely to play a role in the sheltering of the site closest to the glacier margin, especially considering the sharp slope transitions and short distances involved. Thus, the flow hitting the glacier edge may not be well developed and still be affected by lee-side flow separation etc, reducing its ability to influence the stable glacier boundary layer. This may be worth discussing further here.

**Response:** we thank Jono Conway for his thoughtful comments and revised the conclusion now reading: The topographic setting which is typical for alpine glaciers are likely to play a significant role in the sheltering of the site closest to the glacier margin. Steep moraine sides and sharp slope transitions at the glacier margin strongly affect the local boundary layer flow (i.e. lee-side flow separation) reducing the ability of the flow hitting the glacier edge to influence the stable glacier boundary layer. Contrary, well developed flows at the glacier line and associated higher wind speeds appear to promote turbulent mixing close to the surface allowing the rush-in of high-speed fluid from the outer region into the near-surface atmospheric layer, as shown by Mott et al., (2016) for a wind tunnel experiment with warm air advection over a melting snow surface.

• 528 – as the study only presents data from 5 days, it would be more meaningful to say "during five days that displayed a distinct disruption of down-glacier flow during a three-week period in summer 2018." Or similar.

**Response**: we followed this suggestion and revised this part of the manuscript.

• 541 – 'induced by strong westerly winds' – while this makes sense, the origin of the flow is still speculative so please revise.

**Response:** we revised this paragraph not speculating about the origin of the flow.

• 552 – 'At the peripheral areas stronger exposure' – shouldn't this be 'weaker exposure'. **Response:** Yes, we changed that to weaker exposure.

552 – As wind direction is not presented for TT1 it is impossible to assess if the 'preservation of a very-shallow low-level katabatic jet' is supported by the results. Figure 1 shows the WD is aligned at all levels at TT3 during disturbed situations – in order to support a katabatic jet at TT1 the wind direction would need to be maintained down-slope. The BL could still be decoupled at TT1 because of the strong thermal stratification, but this does not necessarily mean that a katabatic jet will exist at TT1. Please revise.

**Response**: Yes, the referee is right at this point and we try to be more clear that the turbulence data (positive momentum fluxes) indicate a wind jet below the lowest measurement level but data do not allow to distinguish between a glacier flow or slope flow: At the peripheral area weaker exposure to the westerly winds might promote the preservation of a very shallow low-level jet which potentially decouples near-surface turbulence from higher atmospheric levels (Parmhed et al., 2004). Although no wind direction measurements are available at heights below 1.7 m, positive momentum fluxes at the lowest measurement height indicate the existence of such a shallow low-level jet height which might be connected to a glacier flow or a thermal flow originating from the moraine slopes.

• 575 – "the frequency of such flows at other glaciers is not known" – this comment highlights that fact that the frequency of these flows has not been presented in the current study. This would be an easy and useful addition to the results.

**Response**: During the entire 3 weeks of data 20 % of the data fulfilled the conditions of disturbed conditions. 45% of the data is categorized as katabatic conditions. We added this information to the method section.

**Editorial comments:**

• Temporal changes

**Response**: we changed change to changes.

• 121 – 'while as the fifth tower (WT1), with at these' -> 'while at the fifth tower (WT!), these' **Response:** thanks, we revised this.

• 125 – suggest changing 'methodology' to 'data processing'

**Response**: we changed methodology to data processing and also change turbulence towers to instrumentation

Figure 2: Multi-resolution flux decomposition of buoyancy flux as a function of time scale t for the four examined stations.

---

## Referee Report (RR1)

I would like to thank the authors for their careful consideration of all comments made. The paper is well written and all discussion points are now very clearly substantiated by the figures provided.

My only remaining concern is clarity in Figure 4 and Figure 6. Currently, it's difficult to differentiate between days due to overlapping colours and colours of different shades – especially in c/d. If the aim is to highlight the days, either more subplots or mean profiles might help with clarity. However, as it is currently difficult to distinguish between days, changing all to the same colour would likely capture the same information (possibly still including a mean profile?)

Some minor editorial comments:
- The units of momentum flux in Fig 6 need to be revised.
- The overbars on Fig 6 and on other would look better if they spanned the width of a'b'
- It is my preference for clarity in multiplots to have the distributions on the top and right of each scatter. Then they can lie flush with the plots and not have axis labels in between.
- The whitespace on Table 2 could be reduced.
- On Table 3, I assume that d1,d2… indicates days, but this notation isn't used elsewhere, nor is it explicitly defined.

---

## Author Response (AR2)

**Response to Referee 1:**

We thank referee 1 for his/her comments. We provide here our responses to those comments and describe how we addressed them in the revised manuscript. The original reviewer comments are in normal black font while our answers appear in blue font.

My only remaining concern is clarity in Figure 4 and Figure 6. Currently, it's difficult to differentiate between days due to overlapping colours and colours of different shades – especially in c/d. If the aim is to highlight the days, either more subplots or mean profiles might help with clarity. However, as it is currently difficult to distinguish between days, changing all to the same colour would likely capture the same information (possibly still including a mean profile?)
We revised Figures 4 and 6 according to the referee comment. We decided to show all lines in black and added averaged profiles for each measurement day with the same colours as shown in Figure 3.

Some minor editorial comments:
- The units of momentum flux in Fig 6 need to be revised. We revised the unit of momentum flux in Figure 6 and elsewhere.
- The overbars on Fig 6 and on other would look better if they spanned the width of a'b'. We changed the overbars in all figures. Now spanning the width of a'b'.
- It is my preference for clarity in multiplots to have the distributions on the top and right of each scatter. Then they can lie flush with the plots and not have axis labels in between. We think that this is kind of personal preference and feel more comfortable in the way the distributions are shown now.
- The whitespace on Table 2 could be reduced. We revised all Tables not showing any whitespaces anymore. We now use shading to indicate whether the data is classified as katabatic or disturbed condition.
- On Table 3, I assume that d1,d2… indicates days, but this notation isn't used elsewhere, nor is it explicitly defined. We revised the chapter of Table 3 accordingly.